# Optimistic Acceleration for Optimization

## Abstract

We consider new variants of optimization algorithms. Our algorithms are based on the observation that mini-batch of stochastic gradients in consecutive iterations do not change drastically and consequently may be predictable. Inspired by the similar setting in online learning literature called Optimistic Online learning, we propose two new optimistic algorithms for AMSGrad and Adam, respectively, by exploiting the predictability of gradients. The new algorithms combine the idea of momentum method, adaptive gradient method, and algorithms in Optimistic Online learning, which leads to speed up in training deep neural nets in practice.

## 1 Introduction

Nowadays deep learning has been shown to be very effective in several tasks, from robotics (e.g. Levine et al. (2017)), computer vision (e.g. He et al. (2016); Goodfellow et al. (2014)), reinforcement learning (e.g. Mnih et al. (2013), to natural language processing (e.g. Graves et al. (2013)). Typically, the model parameters of a state-of-the-art deep neural net is very high-dimensional and the required training data is also in huge size. Therefore, fast algorithms are necessary for training a deep neural net. To achieve this, there are number of algorithms proposed in recent years, such as AMSGrad (Reddi et al. (2018)), Adam (Kingma & Ba (2015)), RMSProp (Tieleman & Hinton (2012)), AdaDelta (Zeiler (2012)), and Nadam (Dozat (2016)), etc.

All the prevalent algorithms for training deep nets mentioned above combines two ideas: the idea of adaptivity in AdaGrad (Duchi et al. (2011); McMahan & Streeter (2010)) and the idea of momentum as Nesterov's Method (Nesterov (2004)) or the Heavy ball method (Polyak (1964)). AdaGrad is an online learning algorithm that works well compared to the standard online gradient descent when the gradient is sparse. The update of AdaGrad has a notable feature: the learning rate is different for different dimensions, depending on the magnitude of gradient in each dimension, which might help in exploiting the geometry of data and leading to a better update. On the other hand, Nesterov's Method or the Momentum Method (Polyak (1964)) is an accelerated optimization algorithm whose update not only depends on the current iterate and current gradient but also depends on the past gradients (i.e. momentum). State-of-the-art algorithms like AMSGrad (Reddi et al. (2018)) and Adam (Kingma & Ba (2015)) leverages these two ideas to get fast training for neural nets.

In this paper, we propose an algorithm that goes further than the hybrid of the adaptivity and momentum approach. Our algorithm is inspired by Optimistic Online learning (Chiang et al. (2012); Rakhlin & Sridharan (2013); Syrgkanis et al. (2015); Abernethy et al. (2018)). Optimistic Online learning considers that a good guess of the loss function in the current round of online learning is available and plays an action by exploiting the good guess. By exploiting the guess, those algorithms in Optimistic Online learning have regret in the form of $O(\sqrt{\sum_{t=1}^{T} \|g_t - m_t\|})$, where $g_t$ is the gradient of loss function in round $t$ and $m_t$ is the "guess" of $g_t$ before seeing the loss function in round $t$ (i.e. before getting $g_t$). This kind of regret can be much smaller than $O(\sqrt{T})$ when one has a good guess $m_t$ of $g_t$. We combine the Optimistic Online learning idea with the adaptivity and the momentum ideas to design new algorithms in training deep neural nets, which leads to New-Optimistic-AMSGrad and new-optimistic-adam. We also provide theoretical analysis of New-Optimistic-AMSGrad. The proposed Optimistic- algorithms not only adapt to the informative dimensions and exhibit momentums but also take advantage of a good guess of the next gradient to facilitate acceleration. We evaluate our algorithms with (Kingma & Ba (2015)), (Reddi et al. (2018)) and (Daskalakis et al. (2018)). Experiments show that our Optimistic-algorithms are faster than the baselines. We should explain that Daskalakis et al. (2018) proposed another version of optimistic algorithm for Adam, which is referred to as Adam-DISZ in this paper. We apply the idea of (Daskalakis et al. (2018)) on AMSGrad, which leads to AMSGrad-DISZ. Both Adam-DISZ and AMSGrad-DISZ are used as baselines.

## 2 Preliminaries

Both AMSGrad (Reddi et al. (2018)) and Adam (Kingma & Ba (2015)) are actually Online learning algorithms. They use Regret analysis to provide some theoretical guarantees of the algorithms. Since one can convert an online learning algorithm to an offline optimization algorithm by online-to-batch conversion (Cesa-Bianchi et al.

(2004)), one can design an offline optimization algorithm by designing and analyzing its counterpart in online learning. Therefore, we would like to give a brief review of ONLINE LEARNING and OPTIMISTIC-ONLINE LEARNING.

## 2.1 BRIEF REVIEW OF ONLINE LEARNING

In the typical setting of online learning, there is a LEARNER playing an action and then receiving a loss function in each round $t$. Specifically, the learner plays an action $w_t \in \mathcal{K}$ in round $t$, where $w_t$ is chosen in a compact and convex set $\mathcal{K} \subseteq \mathbb{R}^n$, known as the DECISION SPACE. Then, the learner sees the LOSS FUNCTION $\ell_t(\cdot)$ and suffers loss $\ell_t(w_t)$ for the choice. No distributional assumption is made on the loss functions sequence $\{\ell_1(\cdot), \ell_2(\cdot), \ldots, \ell_T(\cdot)\}$ in ONLINE LEARNING. Namely, the loss functions can be adversarial. The goal of an online learner is minimizing its REGRET, which is

$$\text{Regret}_T := \sum_{t=1}^{T} \ell_t(w_t) - \min_{w \in \mathcal{K}} \sum_{t=1}^{T} \ell_t(w). \tag{1}$$

We can also define AVERAGE REGRET as $\overline{\text{Regret}}_T := \frac{\text{Regret}_T}{T}$, which is REGRET divided by number of rounds $T$. In ONLINE LEARNING literature, NO-REGRET ALGORITHMS means online learning algorithms satisfying $\overline{\text{Regret}}_T \to 0$ as $T \to \infty$.

## 2.2 BRIEF REVIEW OF OPTIMISTIC ONLINE LEARNING

In recent years, there is a branch of works in the paradigm of OPTIMISTIC ONLINE LEARNING (e.g. Chiang et al. (2012); Rakhlin & Sridharan (2013); Syrgkanis et al. (2015); Abernethy et al. (2018)). The idea of OPTIMISTIC ONLINE LEARNING is as follows. Suppose that, in each round $t$, the learner has a good guess $m_t(\cdot)$ of the loss function $\ell_t(\cdot)$ before playing an action $w_t$. (Recall that the learner receives the loss function after the learner commits an action!) Then, the learner should exploit the guess $m_t(\cdot)$ to choose an action $w_t$, as $m_t(\cdot)$ is close to the true loss function $\ell_t(\cdot)$. [1]

For example, (Syrgkanis et al. (2015)) proposed an optimistic-variant of the so called FOLLOW-THE-REGULARIZED-LEADER (FTRL). The update rule of FTRL (see e.g. Hazan (2016)) is

$$w_t = \arg\min_{w \in \mathcal{K}} \langle w, L_{t-1} \rangle + \frac{1}{\eta} R(w), \tag{2}$$

where $R(\cdot)$ is a 1-strong convex function with respect to a norm ($\|\cdot\|$) on the constraint set $\mathcal{K}$, $L_{t-1} := \sum_{s=1}^{t-1} g_s$ is the cumulative sum of gradient vectors of the convex loss functions (i.e. $g_s := \nabla \ell_s(w_s)$) up to but not including $t$, and $\eta$ is a parameter. The OPTIMISTIC-FTRL of (Syrgkanis et al. (2015)) has update

$$w_t = \arg\min_{w \in \mathcal{K}} \langle w, L_{t-1} + m_t \rangle + \frac{1}{\eta} R(w), \tag{3}$$

where $m_t$ is the learner's guess of the gradient vector $g_t := \nabla \ell_t(w_t)$. Under the assumption that loss functions are convex, the regret of OPTIMISTIC-FTRL satisfies $\text{Regret}_T \leq O(\sqrt{\sum_{t=1}^{T} \|g_t - m_t\|_*})$, which can be much smaller than $O(\sqrt{T})$ of FTRL if $m_t$ is close to $g_t$. Consequently, OPTIMISTIC-FTRL will have much smaller regret than FTRL. On the other hand, if $m_t$ is far from $g_t$, then the regret of OPTIMISTIC-FTRL would be a constant factor worse than that of FTRL without optimistic update. In the later section, we provide a way to get $m_t$. Here, we just want to emphasize the importance of leveraging a good guess $m_t$ for updating $w_t$ to get a fast convergence rate (or equivalently, small regret). We also note that the works of OPTIMISTIC ONLINE LEARNING Chiang et al. (2012); Rakhlin & Sridharan (2013); Syrgkanis et al. (2015)) has been shown to accelerate the convergence of some zero-sum games.

## 2.3 ADAM AND AMSGRAD

ADAM (Kingma & Ba (2015)) is a very popular algorithm for training deep nets. It combines the momentum idea Polyak (1964) with the idea of ADAGRAD (Duchi et al. (2011)), which has individual learning rate for different dimensions. The learning rate of ADAGRAD in iteration $t$ for a dimension $j$ is proportional to the inverse of $\sqrt{\Sigma_{s=1}^{t} g_s[j]^2}$, where $g_s[j]$ is the $j_{th}$ element of the gradient vector $g_s$ in time $s$. This adaptive learning rate may help for accelerating the convergence when the gradient vector is sparse (Duchi et al. (2011)). However, when applying ADAGRAD to train deep nets, it is observed that the learning rate might decay too fast (Kingma & Ba (2015)). Therefore, (Kingma &

---

[1] Imagine that if the learner would had been known $\ell_t(\cdot)$ before committing its action, then it would exploit the knowledge to determine its action and minimizes the regret.

---

**Algorithm 1** AMSGRAD (Reddi et al. (2018))

1: Required: parameter $\beta_1$, $\beta_2$, and $\eta_t$.
2: Init: $w_1$.
3: **for** $t = 1$ to $T$ **do**
4:     Get mini-batch stochastic gradient vector $g_t \in \mathbb{R}^d$ at $w_t$.
5:     $\theta_t = \beta_1 \theta_{t-1} + (1 - \beta_1) g_t$.
6:     $v_t = \beta_2 v_{t-1} + (1 - \beta_2) g_t^2$.
7:     $\hat{v}_t = \max(\hat{v}_{t-1}, v_t)$.
8:     $w_{t+1} = w_t - \eta_t \frac{\theta_t}{\sqrt{\hat{v}_t}}$.  (element-wise division)
9: **end for**

---

Ba (2015)) proposes using a moving average of gradients (element-wise) divided by the root of the second moment of the moving average to update model parameter $w$ (i.e. line 5,6 and line 8 of Algorithm 1). Yet, ADAM (Kingma & Ba (2015)) fails at some online convex optmization problems. AMSGRAD (Reddi et al. (2018)) fixes the issue. The algorithm of AMSGRAD is shown in Algorithm 1. The difference between ADAM and AMSGRAD lies on line 7 of Algorithm 1. ADAM does not have the update of line 7. (Reddi et al. (2018)) adds the step to guarantee a non-increasing learning rate, $\frac{\eta_t}{\sqrt{\hat{v}_t}}$. which helps for the convergence (i.e. average regret $\overline{\text{Regret}}_T \to 0$.) For the parameters of AMSGRAD, it is suggested that $\beta_1 = 0.9$, $\beta_2 = 0.99$ and $\eta_t = \eta/\sqrt{t}$ for a number $\eta$.

## 3 NEW-OPTIMISTIC-AMSGRAD

As mentioned in the introduction, Daskalakis et al. (2018) proposed one version of optimistic algorithm for ADAM, which is referred to as ADAM-DISZ in this paper. Daskalakis et al. (2018) did not propose an optimistic algorithm for AMSGRAD but such an extension is straightforward which is referred to as AMSGRAD-DISZ.

In this section, we propose a new algorithm for training deep nets: NEW-OPTIMISTIC-AMSGRAD, shown in Algorithm 2. NEW-OPTIMISTIC-AMSGRAD has an optimistic update, which is line 9 of Algorithm 2. It exploits the guess $m_{t+1}$ of $g_{t+1}$ to get $w_{t+1}$, since the vector $h_{t+1}$ uses $m_{t+1}$. Notice that the gradient vector is computed at $w_t$ instead of $w_{t-\frac{1}{2}}$ and the moving average of gradients is used to update $w_{t+\frac{1}{2}}$. One might want to combining line 8 and line 9 and getting a single line: $w_{t+1} = w_{t-\frac{1}{2}} - \eta_t \frac{\theta_t}{\sqrt{\hat{v}_t}} - \eta_{t+1} \frac{4}{1-\beta_1} \frac{h_{t+1}}{\sqrt{\hat{v}_t}}$. From this, we see that $w_{t+1}$ is updated from $w_{t-\frac{1}{2}}$ instead of $w_t$. Therefore, while NEW-OPTIMISTIC-AMSGRAD looks like just doing an additional update compared to AMSGRAD, the difference of update is subtle. We also want to emphasize that although the learning rate on line 9 contains $\frac{4}{1-\beta_1}$ factor. We suspect that it is due to the artifact of our theoretical analysis. In our experiments, the learning rate on line 9 does not have the factor of $\frac{4}{1-\beta_1}$. That is, in practice, we implement line 9 as $w_{t+1} = w_{t+\frac{1}{2}} - \eta_{t+1} \frac{h_{t+1}}{\sqrt{\hat{v}_t}}$. We leave closing the gap between theory and practice as a future work.

We see that NEW-OPTIMISTIC-AMSGRAD inherits three properties

- Adaptive learning rate of each dimension as ADAGRAD (Duchi et al. (2011)). (line 8)
- Exponentially moving average of the past gradients as NESTEROV'S METHOD (Nesterov (2004)) and the HEAVY-BALL method (Polyak (1964)). (line 5)
- Optimistic update that exploits a good guess of the next gradient vector as optimistic online learning algorithms (Chiang et al. (2012); Rakhlin & Sridharan (2013); Syrgkanis et al. (2015); Abernethy et al. (2018)). (line 9)

The first property helps acceleration when the gradient has sparse structure. The second one is the well-recognized idea of momentum which can achieve acceleration. The last one, perhaps less known outside the ONLINE LEARNING community, can actually achieve acceleration when the prediction of the next gradient is good. We are going to elaborate this property in the later section where we give the theoretical analysis of NEW-OPTIMISTIC-AMSGRAD.

**To obtain** $m_t$, we use the extrapolation algorithm of (Scieur et al. (2016)). Extrapolation studies estimating the limit of sequence using the last few iterates (Brezinski & Zaglia (2013)). Some classical works include Anderson acceleration (Walker & Ni. (2011)), minimal polynomial extrapolation (Cabay & Jackson (1976)), reduced rank extrapolation (Eddy (1979)). These method typically assumes that the sequence $\{x_t\} \in \mathbb{R}^d$ has a linear relation

$$x_t = A(x_{t-1} - x^*) + x^*, \tag{4}$$

for an unknown matrix $A \in \mathbb{R}^{d \times d}$ (not necessarily symmetric). The goal is to use the last few iterates $\{x_t\}$ to estimate the fixed point $x^*$ on (4). (Scieur et al. (2016)) adapt the classical extrapolation methods to the iterates/updates of

---

**Algorithm 2** NEW-OPTIMISTIC-AMSGRAD

---

1: Required: parameter $\beta_1$, $\beta_2$, and $\eta_t$.
2: Init: $w_1$.
3: **for** $t = 1$ to $T$ **do**
4:     Get mini-batch stochastic gradient vector $g_t \in \mathbb{R}^d$ at $w_t$.
5:     $\theta_t = \beta_1 \theta_{t-1} + (1 - \beta_1) g_t$.
6:     $v_t = \beta_2 v_{t-1} + (1 - \beta_2) g_t^2$.
7:     $\hat{v}_t = \max(\hat{v}_{t-1}, v_t)$.
8:     $w_{t+\frac{1}{2}} = w_{t-\frac{1}{2}} - \eta_t \frac{\theta_t}{\sqrt{\hat{v}_t}}$.  (element-wise division)
9:     $w_{t+1} = w_{t+\frac{1}{2}} - \eta_{t+1} \frac{4}{1-\beta_1} \frac{h_{t+1}}{\sqrt{\hat{v}_t}}$, where $h_{t+1} := \beta_1 \theta_{t-1} + (1 - \beta_1) m_{t+1}$
        and $m_{t+1}$ is the guess of $g_{t+1}$. (In practice, use $w_{t+1} = w_{t+\frac{1}{2}} - \eta_{t+1} \frac{h_{t+1}}{\sqrt{\hat{v}_t}}$.)
10: **end for**

---

**Algorithm 3** REGULARIZED APPROXIMATE MINIMAL POLYNOMIAL EXTRAPOLATION (RMPE) (Scieur et al. (2016))

---

1: Input: some sequence $\{x_s \in \mathbb{R}^d\}_{s=0}^{s=r}$, parameter $\lambda > 0$
2:     Compute matrix $U = [x_1 - x_0, \ldots, x_r - x_{r-1}] \in \mathbb{R}^{d \times r}$
3:     Obtain $z$ by solving $(U^\top U + \lambda I)z = \mathbf{1}$.
4:     Get $c = z/(z^\top \mathbf{1})$
5: Output: $\Sigma_{i=0}^{r-1} c_i x_i$, the approximation of the fixed point $x^*$.

---

an optimization algorithm and propose an algorithm that produces a solution that is better than the last iterate of the underlying optimization algorithm in practice. The algorithm of (Scieur et al. (2016)) (shown in Algorithm 3) allows the iterates $\{x_t\}$ to be nonlinear

$$x_t - x^* = A(x_{t-1} - x^*) + e_t, \tag{5}$$

where $e_t$ is a second order term (namely, satisfying $\|e_t\|_2 = O(\|x_{t-1} - x^*\|_2^2)$). Some theoretical guarantees regarding the distance between the output and $x^*$ is provided in (Scieur et al. (2016)).

In NEW-OPTIMISTIC-AMSGRAD, we use Algorithm 3 to get $m_t$. Specifically, $m_t$ is obtained by

- Call Algorithm 3 with input being a sequence of some past $r + 1$ gradients, $\{g_t, g_{t-1}, g_{t-2}, \ldots, g_{t-r}\}$ to obtain $m_t$, where $r$ is a parameter.

- Set $m_t := \Sigma_{i=0}^{r-1} c_i g_{t-r+i}$ from the output of Algorithm 3.

If the past few gradients can be modeled by (4) approximately, the extrapolation method should be expected to work well in predicting the gradient. In practice, it helps to achieve faster convergence.

**NEW-OPTIMISTIC-ADAM** By removing line 7 in Algorithm 2, the step of making monotone weighted second moment, we obtain an algorithm which we call it NEW-OPTIMISTIC-ADAM, as the resulting algorithm can be viewed as an OPTIMISTIC-variant of ADAM.

### 3.1 THEORETICAL ANALYSIS OF NEW-OPTIMISTIC-AMSGRAD

We provide the regret analysis here. We denote the Mahalanobis norm $\| \cdot \|_H = \sqrt{\langle \cdot, H \cdot \rangle}$ for some PSD matrix $H$. For the PSD matrix $\text{diag}\{\hat{v}_t\}$, where $\text{diag}\{\hat{v}_t\}$ represents the diagonal matrix such that its $i_{th}$ diagonal element is $\hat{v}_t[i]$ in Algorithm 2, we define the the corresponding Mahalanobis norm $\| \cdot \|_{\psi_t} := \sqrt{\langle \cdot, \text{diag}\{\hat{v}_t\}^{1/2} \cdot \rangle}$, where we use the notation $\psi_t$ to represent the matrix $\text{diag}\{\hat{v}_t\}^{1/2}$. We can also define the the corresponding dual norm $\| \cdot \|_{\psi_t^*} := \sqrt{\langle \cdot, \text{diag}\{\hat{v}_t\}^{-1/2} \cdot \rangle}$.

We assume that the model parameter $w$ is in $d$-dimensional space. That is, $w \in \mathbb{R}^d$. Also, the analysis of NEW-OPTIMISTIC-AMSGRAD is for unconstrained optimization. Thus, we assume that the constraint $\mathcal{K}$ of the benchmark in the regret definition, $\min_{w \in \mathcal{K}} \sum_{t=1}^T \ell_t(w)$, is a finite norm ball that contains the optimal solutions to the underlying offline unconstrained optimization problem.

Now we can conduct our analysis. First of all, we can decompose the regret as follows.

$$\text{Regret}_T := \sum_{t=1}^{T} \ell_t(w_t) - \min_{w \in \mathcal{K}} \sum_{t=1}^{T} \ell_t(w) \leq \sum_{t=1}^{T} \langle w_t - w^*, \nabla \ell_t(w_t) \rangle := \sum_{t=1}^{T} \langle w_t - w^*, g_t \rangle$$

$$= \sum_{t=1}^{T} \langle w_{t-\frac{1}{2}} - w^*, g_t \rangle + \sum_{t=1}^{T} \langle w_t - w_{t-\frac{1}{2}}, g_t \rangle \tag{6}$$

$$= \sum_{t=1}^{T} \langle w_{t-\frac{1}{2}} - w^*, g_t \rangle + \sum_{t=1}^{T} \langle w_t - w_{t-\frac{1}{2}}, g_t - h_t \rangle + \sum_{t=1}^{T} \langle w_t - w_{t-\frac{1}{2}}, h_t \rangle.$$

where the first inequality is by assuming that the loss function $\ell_t(\cdot)$ is convex and that we use the notation $g_t := \nabla \ell_t(w_t)$ which we adopt throughout the following proof for brevity.

Given the decomposition, let us analyze the first term $\sum_{t=1}^{T} \langle w_{t-\frac{1}{2}} - w^*, g_t \rangle$ in (6).

**Lemma 1.** *Denote $D_\infty = \max_t \|w_{t-\frac{1}{2}} - w^*\|_\infty$. We have that $\sum_{t=1}^{T} \langle w_{t-\frac{1}{2}} - w^*, g_t \rangle \leq \frac{1}{2\eta_T(1-\beta_1)} D_\infty^2 \sum_{i=1}^{d} \hat{v}_T[i]^2 + \sum_{t=1}^{T} \frac{\eta_t}{(1-\beta_1)} \|\theta_t\|_{\psi_t^*}^2 + D_\infty^2 \sum_{t=1}^{T} \sum_{i=1}^{d} \frac{\beta_1 \hat{v}_t[i]^{1/2}}{2\eta_t(1-\beta_1)}.$*

Astute readers may realize that the bound in Lemma 1 is actually the bound of AMSGRAD. Indeed, since in online learning setting the loss vectors $g_t$ come adversarially, it does matter how $g_t$ is generated. Therefore, the regret of $\sum_{t=1}^{T} \langle w_{t-\frac{1}{2}} - w^*, g_t \rangle$ can be bounded in the same way as AMSGRAD. In Appendix B, we provide the detail proof of Lemma 1.

Now we switch to bound the other sums $\sum_{t=1}^{T} \langle w_t - w_{t-\frac{1}{2}}, g_t - h_t \rangle + \sum_{t=1}^{T} \langle w_t - w_{t-\frac{1}{2}}, h_t \rangle$ in (6). The proof is available in Appendix C.

**Lemma 2.** $\sum_{t=1}^{T} \langle w_t - w_{t-\frac{1}{2}}, g_t - h_t \rangle + \sum_{t=1}^{T} \langle w_t - w_{t-\frac{1}{2}}, h_t \rangle \leq \sum_{t=1}^{T} \frac{\eta_t}{2} \|g_t - h_t\|_{(\frac{1-\beta_1}{8} \psi_{t-1})^*}^2 - \frac{3}{2\eta_t} \|w_t - w_{t-\frac{1}{2}}\|_{\frac{1-\beta_1}{8} \psi_{t-1}}^2.$

Combining (6) and Lemma 1 and 2 leads to

$$\text{Regret}_T \leq \frac{1}{2\eta_T(1-\beta_1)} D_\infty^2 \sum_{i=1}^{d} \hat{v}_T[i]^2 + \sum_{t=1}^{T} \frac{\eta_t}{(1-\beta_1)} \|\theta_t\|_{\psi_t^*}^2$$

$$+ D_\infty^2 \sum_{t=1}^{T} \sum_{i=1}^{d} \frac{\beta_1 \hat{v}_t[i]^{1/2}}{2\eta_t(1-\beta_1)} + \sum_{t=1}^{T} \frac{\eta_t}{2} \|g_t - h_t\|_{(\frac{1-\beta_1}{8} \psi_{t-1})^*}^2 - \frac{3}{2\eta_t} \|w_t - w_{t-\frac{1}{2}}\|_{\frac{1-\beta_1}{8} \psi_{t-1}}^2. \tag{7}$$

Now we can conclude the following theorem. The proof is in Appendix D.

**Theorem 1.** *Denote $\gamma := \beta_1/\sqrt{\beta_2} < 1$ and $D_\infty = \max_t \|w_{t-\frac{1}{2}} - w^*\|_\infty$. Then,*

$$Regret_T \leq \frac{1}{2\eta_T(1-\beta_1)} D_\infty^2 \sum_{i=1}^{d} \hat{v}_T[i]^2 + D_\infty^2 \sum_{t=1}^{T} \sum_{i=1}^{d} \frac{\beta_1 \hat{v}_t[i]^{1/2}}{2\eta_t(1-\beta_1)}$$

$$+ \sum_{t=1}^{T} 2\eta_t(1-\beta_1) \|g_t - m_t\|_{\psi_t^*}^2 + \sum_{t=1}^{T} \frac{\eta_t}{2} \|g_t - h_t\|_{(\frac{1-\beta_1}{8} \psi_{t-1})^*}^2. \tag{8}$$

One should compare the bound with that of AMSGRAD (Reddi et al. (2018)), which is

$$\text{Regret}_T \leq \frac{1}{2\eta_T(1-\beta_1)} D_\infty^2 \sum_{i=1}^{d} \hat{v}_T[i]^2 + D_\infty^2 \sum_{t=1}^{T} \sum_{i=1}^{d} \frac{\beta_1 \hat{v}_t[i]^{1/2}}{2\eta_t(1-\beta_1)}$$

$$+ \frac{\eta\sqrt{1+\log T}}{(1-\beta_1)^2(1-\gamma)\sqrt{1-\beta_2}} \sum_{i=1}^{d} \|g_{1:T}[i]\|_2, \tag{9}$$

where $\eta_t = \eta/\sqrt{t}$ in their setting. We need to compare the last two terms in (8) with the last term in (9). We are going to show that, under certain conditions, the bound is smaller than that of AMSGRAD. Let us suppose that $g_t$ is

close to $m_t$ so that $\sum_{t=1}^{T} 2\eta_t(1-\beta_1)\|g_t - m_t\|_{\psi_t^*}^2$ is much smaller than the last term of (8). Yet, the last term of (8), $\sum_{t=1}^{T} \frac{\eta_t}{2}\|g_t - h_t\|_{(\frac{1-\beta_1}{8}\psi_{t-1})^*}$, might be actually $o(\sqrt{T})$ and consequently might also be smaller than the last term of (9). To see this, let us rewrite $\sum_{t=1}^{T} \frac{\eta_t}{2}\|g_t - h_t\|_{(\frac{1-\beta_1}{8}\psi_{t-1})^*}$ and set $\eta_t = \eta/\sqrt{t}$ as (Reddi et al. (2018)); it is

$$O(\sum_{t=1}^{T}\sum_{i=1}^{d} \eta_t \frac{(g_t[i]^2 - h_t[i]^2)}{\sqrt{\hat{v}_{t-1}[i]}}) = O(\sum_{t=1}^{T}\sum_{i=1}^{d} \frac{\eta}{\sqrt{t}} \frac{(g_t[i]^2 - h_t[i]^2)}{\sqrt{\hat{v}_{t-1}[i]}}). \tag{10}$$

Assume that if each $\frac{(g_t[i]^2 - h_t[i]^2)}{\sqrt{\hat{v}_{t-1}[i]}}$ in the inner sum is bounded by a constant $c$, we will get $\sum_{t=1}^{T} O(\frac{1}{\sqrt{t}}) = O(\sqrt{T})$. Yet, the denominator $\sqrt{\hat{v}_{t-1}}$ is non-decreasing so that we can actually have a smaller bound. That is, in practice, $\sqrt{\hat{v}_{t-1}[i]}$ might grow over time, and the growth rate is different for different dimension $i$. If $\sqrt{\hat{v}_{t-1}[i]}$ in the denominator grows like $O(\sqrt{t})$ then the last term is just $O(\log T)$, which might be better than that of the last term on (9). One can also get a data dependent bound of the last term of (8). To summarize, when $m_t$ is close to $g_t$, NEW-OPTIMISTIC-AMSGRAD can have a smaller regret (thus better convergence rate) than ADAM (Kingma & Ba (2015)) and AMSGRAD (Reddi et al. (2018)).

### 3.2 COMPARISON OF ADAM-DISZ IN DASKALAKIS ET AL. (2018)

---
**Algorithm 4** ADAM-DISZ Daskalakis et al. (2018)

---
1: Required: parameter $\beta_1$, $\beta_2$, and $\eta_t$.
2: Init: $w_1 \in \mathcal{K}$.
3: **for** $t = 1$ to $T$ **do**
4:     Get mini-batch stochastic gradient vector $g_t \in \mathbb{R}^d$ at $w_t$.
5:     $\theta_t = \beta_1\theta_{t-1} + (1-\beta_1)g_t$.
6:     $v_t = \beta_2 v_{t-1} + (1-\beta_2)g_t^2$.
7:     $w_{t+1} = \Pi_k[w_t - 2\eta_t\frac{\theta_t}{\sqrt{v_t}} + \eta_t\frac{\theta_{t-1}}{\sqrt{v_{t-1}}}]$.
8: **end for**

---

We are aware of Algorithm 1 in Daskalakis et al. (2018), which was also motivated by OPTIMISTIC ONLINE LEARNING [2]. For comparison, we replicate ADAM-DISZ in Algorithm 4. We are going to describe the differences of the algorithms and the differences of the contributions between our work and their work. First of all, by comparing NEW-OPTIMISTIC-AMSGRAD (Algorithm 2) or NEW-OPTIMISTIC-ADAM (Algorithm 2 without line 7) with ADAM-DISZ (Algorithm 4), we see that the updates are indeed different. The update cannot be written into the same form as our NEW-OPTIMISTIC-AMSGRAD (and vise versa). NEW-OPTIMISTIC-AMSGRAD (Algorithm 2) actually uses two interleaving sequences of updates $\{w_t\}_{t=1}^T$, $\{w_{t-\frac{1}{2}}\}_{t=1}^T$, Also, Daskalakis et al. (2018) does not have any theoretical analysis of ADAM-DISZ. The theoretical analysis in Daskalakis et al. (2018) is for other algorithms. Second, the contributions are very different. Daskalakis et al. (2018) focus on training GANs Goodfellow et al. (2014). GANs is a two-player zero-sum game. There have been some related works in OPTIMISTIC ONLINE LEARNING like Chiang et al. (2012); Rakhlin & Sridharan (2013); Syrgkanis et al. (2015)) showing that if both players use some kinds of OPTIMISTIC-update, then acceleration to the convergence of the minimax value of the game is possible. Daskalakis et al. (2018) was inspired by these related works and showed that OPTIMISTIC-MIRROR-DESCENT can avoid the cycle behavior in a bilinear zero-sum game, which accelerates the convergence. Our work is about solving $\min_x f(x)$ (e.g. empirical risk) quickly.

We also show that ADAM-DISZ suffers the non-convergence issue as ADAM. The proof is available in Appendix E.

**Theorem 2.** *There exists a convex online learning problem such that* ADAM-DISZ *has nonzero average regret (i.e. $\frac{Regret}{T} \neq o(T)$)*

One might wonder if the non-convergence issue can be avoided if one let the weighted second moment of ADAM-DISZ be monotone by adding the step $\hat{v}_t = \max(\hat{v}_{t-1}, v_t)$ as AMSGRAD, which we call it AMSGRAD-DISZ. Unfortunately, we are unable to prove if the step guarantees convergence or not.

---

[2] We notice that the term NEW-OPTIMISTIC-ADAM was already used in Daskalakis et al. (2018) to describe their Algorithm 1 in the paper. To avoid confusion, let us still use NEW-OPTIMISTIC-ADAM to refer to Algorithm 2 without line 7 in this paper, while we use ADAM-DISZ to refer to their algorithm, where DISZ are the initials of the authors' last names of the paper Daskalakis et al. (2018).

# 4 EXPERIMENTS

To demonstrate the effectiveness of our proposed method, we test its performance with various neural network architectures, including fully-connected neural networks, convolutional neural networks (CNN's) and recurrent neural networks (RNN's). The results illustrate that NEW-OPTIMISTIC-AMSGRAD is able to speed up the convergence of state-of-art AMSGRAD algorithm, making the learning process more efficient.

For AMSGRAD algorithm, we set the parameter $\beta_1$ and $\beta_2$, respectively, to be 0.9 and 0.999, as recommended in Reddi et al. (2018). We tune the learning rate $\eta$ over a fine grid and report the results under best-tuned parameter setting. For NEW-OPTIMISTIC-AMSGRAD and AMSGRAD-DISZ, we use same $\beta_1$, $\beta_2$ and learning rate as those for AMSGRAD to make a fair comparison of the enhancement brought by the optimistic step. We use the same weight initialization for all algorithms. The remaining tuning parameter of NEW-OPTIMISTIC-AMSGRAD is $r$, the number of previous gradients that we use to predict the next move. We conduct NEW-OPTIMISTIC-AMSGRAD with different values of $r$ and observe similar performance (See Appendix A). Hence, we report $r = 15$ for all experiments for tidiness of plots. To follow previous works of Reddi et al. (2018) and Kingma & Ba (2015), we compare different methods on MNIST, CIFAR10 and IMDB datasets in our experiment. For MNIST, we use a noisy version named as MNIST-back-rand in Larochelle et al. (2007) to increase the training difficulty.

## 4.1 MNIST-BACK-RAND + MULTI-LAYER NEURAL NETWORKS

Our experiments start with fully connected neural network for multi-class classification problems. MNIST-back-rand dataset consists of 12000 training samples and 50000 test samples, where random background is inserted in original MNIST hand written digit images. The input dimension is 784 ($28\times28$) and the number of classes is 10. We investigate a multi-layer neural networks with input layer followed by a hidden layer with 200 cells, which is then connected to a layer with 100 neurons before the output layer. All hidden layer cells are rectifier linear units (ReLu's). We use mini-batch size 128 to calculate stochastic gradient in each iteration. Model performance is evaluated by multi-class cross entropy loss. The training loss with respect to number of iterations is reported in Figure 1.

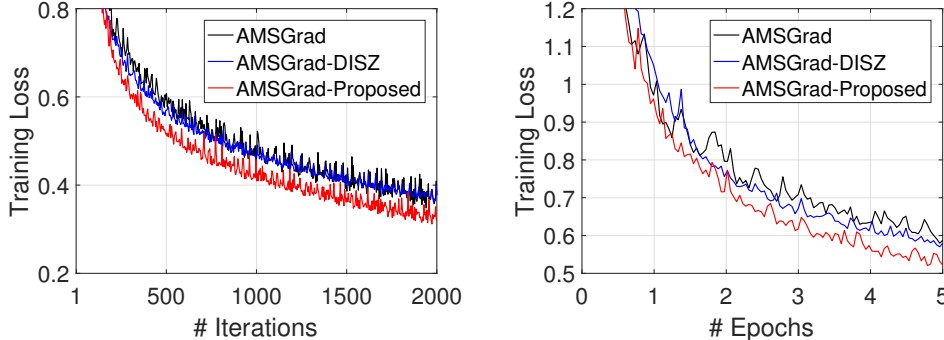

Figure 1: Fully connected neural networks on MNIST-Back-Rand dataset. Left panel: training loss vs. number of iterations. Right panel: training loss only for 5 epochs. One epoch means all training data points are used once. For this dataset, since the training size is 12000 and the mini-batch size is 128, one epoch means 93.75 iterations. This allows us to better visualize the speed in the first few epochs. In large applications, data-loading is often the bottleneck. In the plots, "AMSGrad-Proposed" is the proposed NEW-OPTIMISIC-AMSGRAD.

On this dataset, we empirically observe obvious improvement of NEW-OPTIMISTIC-AMSGRAD in terms of both convergence speed and training loss. On the other hand, AMSGRAD-DISZ performs similarly to AMSGRAD in general.

## 4.2 CIFAR10 + CONVOLUTIONAL NEURAL NETWORKS (CNN)

Convolutional Neural Networks (CNN) have been widely studied and is important in various deep learning applications such as computer vision and natural language processing. We test the effectiveness of NEW-OPTIMISTIC-AMSGRAD in deep CNN's with dropout. We use the CIFAR10 dataset, which includes 60,000 images (50,000 for training and 10,000 for testing) of size $32 \times 32$ in 10 different classes. ALL-CNN architecture proposed in Springenberg et al. (2015) is implemented with two blocks of $3 \times 3$ convolutional filter, $3 \times 3$ convolutional layer with stride 2 and dropout layer with keep probability 0.5. Another block of $3 \times 3$, $1 \times 1$ convolutional layer and a $6 \times 6$ global averaging pooling is added before the output layer. We apply another dropout with keep probability 0.8 on the input layer. The cost function is multi-class cross entropy. The batch size is 128. The images are all whitened. The training loss is provided in figure 2. The result shows that NEW-OPTIMISTIC-AMSGRAD accelerates the learning process significantly

and gives lowest training cost after 10000 iterations. For this dataset, the performance of AMSGRAD-DISZ is worse than original AMSGRAD.

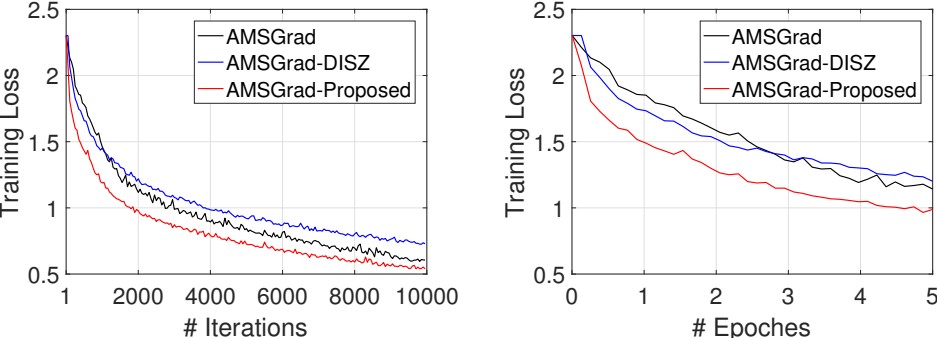

Figure 2: CIFAR10 c96-c96-c96-c192-c192-c192-c192-c192-c10 ConvNet with dropout. Left panel: training loss vs. number of iterations. Right panel: training loss in the first 5 epochs.

### 4.3 IMDB + LONG-SHORT TERM MEMORY (LSTM)

As another important application of deep learning, natural language processing tasks often benefit from considering sequence dependency in the models. Recurrent Neural Networks (RNN's) achieves this goal by adding hidden state units that act as "memory". Long-Short Term Memory (LSTM) is the most popular structure in building RNN's. We use IMDB movie review dataset from Maas et al. (2011) to test the performance of NEW-OPTIMISTIC-AMSGRAD in RNN's under the circumstance of high data sparsity. IMDB is a binary classification dataset with 25000 training and test samples respectively. Our model includes a word embedding layer with 5000 input entries representing most frequent words in the dataset and each word is embedded into a 32 dimensional space. The output of embedding layer is passed to 100 LSTM units, which is then connected to 100 fully connected ReLu's before reaching the output layer. Binary cross-entropy loss is used and the batch size is 128. We provide the results in figure 3.

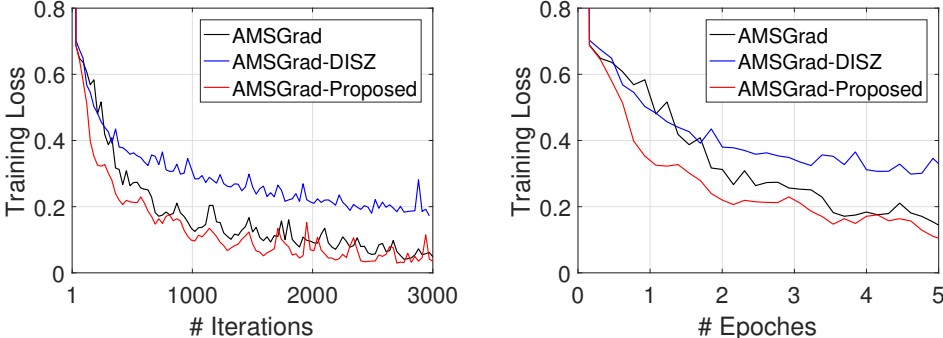

Figure 3: Embedding-LSTM100-f100 RNN: on the left is the training loss against number of iterations. On the right is the plot for first 5 epochs.

We observe a considerable improvement in convergence speed. In the first epoch, the result is more exciting. At epoch 0.5, NEW-OPTIMISTIC-AMSGRAD already achieves the training loss that vanilla AMSGRAD can produce with more than 1 epoch. The sample efficiency is significantly improved. On this dataset, AMSGRAD-DISZ performs less effectively and may be trapped in local minimum. We remark that in each iteration, only a small portion of gradients in embedding layer is non-zero. Thus, this experiment demonstrates that NEW-OPTIMISTIC-AMSGRAD could also perform well with sparse gradient. (**See additional experiments in Appendix A**.)

## 5 EXTENSIONS AND FUTURE WORK

In this paper, we propose NEW-OPTIMISTIC-AMSGRAD, which combines optimistic learning and AMSGRAD to strengthen the learning process of optimization problems, in particular, deep neural networks. The idea of adding optimistic step can be easily extended to other optimization algorithms, $e.g$ ADAM and ADAGRAD. We provide OPTIMISTIC-ADAGRAD algorithm and theoretical results in Appendix F. A potential direction based on this work is to improve the method for predicting next gradient. We expect that optimistic acceleration strategy could be widely used in various optimization problems.

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

## A MORE EXPERIMENTAL RESULTS

### A.1 CHOOSING PARAMETER $r$

In NEW-OPTIMISTIC-AMSGRAD, parameter $r$, the number of previous gradients used, may affect the performance dramatically. If we choose $r$ too small (*e.g* $r < 10$), the optimistic updates will start at very early stage, but the information we collect from the past is limited. This may make our training process off-track at first several iterations. On the other hand, if $r$ is chosen to be too large (*e.g* $r >= 30$), although we may get a better prediction of next gradient, the optimistic step will start late so NEW-OPTIMISTIC-AMSGRAD may miss great chances to improve the learning performance at early stages. Additionally, we need more room to store past gradients, hence the operational time will increase.

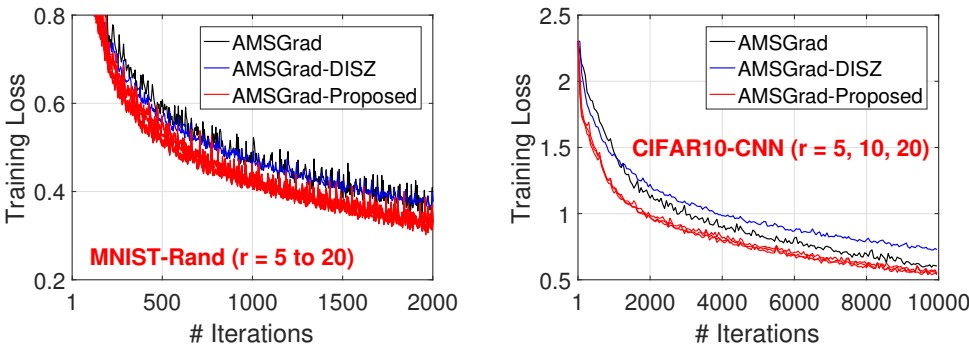

Figure 4: The training performance on two datasets for different $r$ values.

The empirical impact of different $r$ value is reported in figure 4. We suggest $10 \leq r \leq 20$ as an ideal range, and $r = 15$ tend to perform well in most training tasks. Actually, we may make the algorithm more flexible by "early start". For example, when we set $r = 20$, we can instead start adding optimistic step at iteration 10, and gradually increase the number of past gradients we use to predict next move from 10 to 20 in next 10 iterations. After iteration 21, we fix the moving window size for optimistic prediction as 20. This may bring enhancement to NEW-OPTIMISTIC-AMSGRAD because it can seek opportunities to accelerate learning in first several iterations, which is critical for indicating a better direction towards the minimum loss.

## A.2 OPTIMISTIC ADAM

We also conduct experiments on NEW-OPTIMISTIC-ADAM and ADAM-DISZ. The experiment setting is similar to NEW-OPTIMISTIC-AMSGRAD, where we fix $\beta_1$, $\beta_2$ and compare the performance using the best learning rate with respect to ADAM. We provide a brief summary of the results in figure 5, with $r = 15$. The improvement brought by NEW-OPTIMISTIC-ADAM is obvious, indicating that adding optimistic step could also enhance the performance of ADAM optimizer.

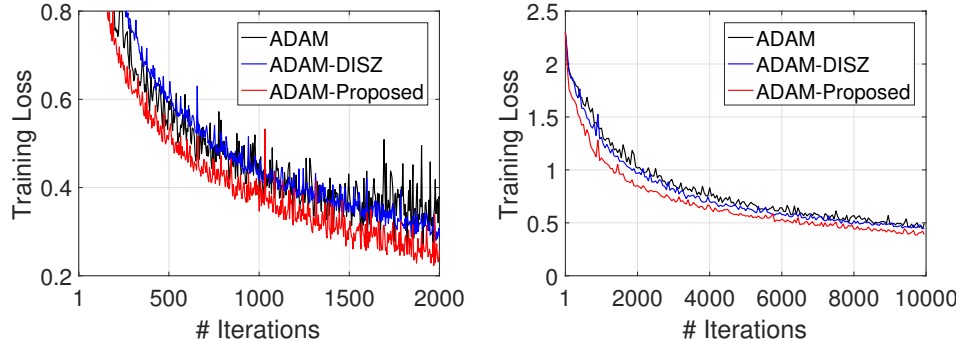

Figure 5: Training loss of NEW-OPTIMISTIC-ADAM. Left panel: MNIST-Back-Rand. Right panel: CIFAR10 (CNN). In the plots, "ADAM-Proposed" is the proposed NEW-OPTIMISIC-ADAM.

## B  PROOF OF LEMMA 1

**Lemma 1**  *Denote* $D_\infty = \max_t \|w_{t-\frac{1}{2}} - w^*\|_\infty$. $\sum_{t=1}^{T} \langle w_{t-\frac{1}{2}} - w^*, g_t \rangle \leq \frac{1}{2\eta_T(1-\beta_1)} D_\infty^2 \sum_{i=1}^{d} \hat{v}_T[i]^2 + \sum_{t=1}^{T} \frac{\eta_t}{(1-\beta_1)} \|\theta_t\|_{\psi_t^*}^2 + D_\infty^2 \sum_{t=1}^{T} \sum_{i=1}^{d} \frac{\beta_1 \hat{v}_t[i]^{1/2}}{2\eta_t(1-\beta_1)}$.

*Proof.*  The proof of this lemma basically follows that of (Reddi et al. (2018)). We have that

$$
\begin{aligned}
\|w_{t+\frac{1}{2}} - w^*\|_{\psi_t}^2 &= \|w_{t-\frac{1}{2}} - \eta_t \text{diag}(\hat{v}_t)^{-1/2}\theta_t - w^*\|_{\psi_t}^2 \\
&= \|w_{t-\frac{1}{2}} - w^*\|_{\psi_t}^2 + \eta_t^2 \|\theta_t\|_{\psi_t^*}^2 - 2\eta_t \langle \theta_t, w_{t-\frac{1}{2}} - w^* \rangle \\
&= \|w_{t-\frac{1}{2}} - w^*\|_{\psi_t}^2 + \eta_t^2 \|\theta_t\|_{\psi_t^*}^2 - 2\eta_t \langle \beta_1 \theta_{t-1} + (1-\beta_1)g_t, w_{t-\frac{1}{2}} - w^* \rangle.
\end{aligned}
\tag{11}
$$

By rearranging the terms above and summing it from $t = 1, \ldots, T$,

$$
\begin{aligned}
\sum_{t=1}^{T} \langle w_{t-\frac{1}{2}} - w^*, g_t \rangle &= \sum_{t=1}^{T} \frac{1}{2\eta_t(1-\beta_1)} \{ \|w_{t-\frac{1}{2}} - w^*\|_{\psi_t}^2 - \|w_{t+\frac{1}{2}} - w^*\|_{\psi_t}^2 \} + \frac{\eta_t}{2(1-\beta_1)} \|\theta_t\|_{\psi_t^*}^2 \\
&\quad + \frac{\beta_1}{1-\beta_1} \langle \theta_{t-1}, w_{t-\frac{1}{2}} - w^* \rangle \\
&\leq \sum_{t=1}^{T} \frac{1}{2\eta_t(1-\beta_1)} \{ \|w_{t-\frac{1}{2}} - w^*\|_{\psi_t}^2 - \|w_{t+\frac{1}{2}} - w^*\|_{\psi_t}^2 \} + \frac{\eta_t}{2(1-\beta_1)} \|\theta_t\|_{\psi_t^*}^2 \\
&\quad + \frac{\beta_1 \eta_t}{2(1-\beta_1)} \|\theta_{t-1}\|_{\psi_t^*}^2 + \frac{\beta_1}{2\eta_t(1-\beta_1)} \|w_{t-\frac{1}{2}} - w^*\|_{\psi_t}^2 \\
&\leq \sum_{t=1}^{T} \frac{1}{2\eta_t(1-\beta_1)} \{ \|w_{t-\frac{1}{2}} - w^*\|_{\psi_t}^2 - \|w_{t+\frac{1}{2}} - w^*\|_{\psi_t}^2 \} + \frac{\eta_t}{(1-\beta_1)} \|\theta_t\|_{\psi_t^*}^2 \\
&\quad + \frac{\beta_1}{2\eta_t(1-\beta_1)} \|w_{t-\frac{1}{2}} - w^*\|_{\psi_t}^2.
\end{aligned}
\tag{12}
$$

where the first inequality is due to Young's inequality and the second one is due to the constraint $0 < \beta_1 \leq 1$ so that $\frac{\eta_t}{2(1-\beta_1)} + \frac{\beta_1 \eta_t}{2(1-\beta_1)} \leq \frac{\eta_t}{(1-\beta_1)}$.

Now we can bound $\sum_{t=1}^{T}\langle w_{t-\frac{1}{2}} - w^*, g_t\rangle$ in (6) as

$$
\begin{aligned}
\sum_{t=1}^{T}\langle w_{t-\frac{1}{2}} - w^*, g_t\rangle &\leq \sum_{t=1}^{T}\frac{1}{2\eta_t(1-\beta_1)}\{\|w_{t-\frac{1}{2}} - w^*\|_{\psi_t}^2 - \|w_{t+\frac{1}{2}} - w^*\|_{\psi_t}^2\} \\
&+ \sum_{t=1}^{T}\frac{\eta_t}{(1-\beta_1)}\|\theta_t\|_{\psi_t^*}^2 + \sum_{t=1}^{T}\frac{\beta_1}{2\eta_t(1-\beta_1)}\|w_{t-\frac{1}{2}} - w^*\|_{\psi_t}^2 \\
&\leq \frac{1}{2\eta_1(1-\beta_1)}\|w_{\frac{1}{2}} - w^*\|_{\psi_1}^2 + \frac{1}{2(1-\beta_1)}\sum_{t=2}^{T}\{\frac{\|w_{t+\frac{1}{2}} - w^*\|_{\psi_t}^2}{\eta_t} - \frac{\|w_{t+\frac{1}{2}} - w^*\|_{\psi_{t-1}}^2}{\eta_{t-1}}\} \\
&+ \sum_{t=1}^{T}\frac{\eta_t}{(1-\beta_1)}\|\theta_t\|_{\psi_t^*}^2 + \sum_{t=1}^{T}\frac{\beta_1}{2\eta_t(1-\beta_1)}\|w_{t-\frac{1}{2}} - w^*\|_{\psi_t}^2 \\
&= \frac{1}{2\eta_1(1-\beta_1)}\sum_{i=1}^{d}\hat{v}_1[i]^2(w_{\frac{1}{2}}[i] - w^*[i])^2 + \frac{1}{2(1-\beta_1)}\sum_{t=2}^{T}\sum_{i=1}^{d}(w_{t-\frac{1}{2}}[i] - w^*[i])^2\{\frac{\hat{v}_t^{1/2}[i]}{\eta_t} - \frac{\hat{v}_{t-1}^{1/2}[i]}{\eta_{t-1}}\} \\
&+ \sum_{t=1}^{T}\frac{\eta_t}{(1-\beta_1)}\|\theta_t\|_{\psi_t^*}^2 + \sum_{t=1}^{T}\frac{\beta_1}{2\eta_t(1-\beta_1)}\|w_{t-\frac{1}{2}} - w^*\|_{\psi_t}^2,
\end{aligned}
\tag{13}
$$

where the first inequality is due to (12).

Continue the analysis, we have

$$
\begin{aligned}
\sum_{t=1}^{T}&\langle w_{t-\frac{1}{2}} - w^*, g_t\rangle \\
&\leq \frac{1}{2\eta_1(1-\beta_1)}\sum_{i=1}^{d}\hat{v}_1[i]^2(w_{\frac{1}{2}}[i] - w^*[i])^2 + \frac{1}{2(1-\beta_1)}\sum_{t=2}^{T}\sum_{i=1}^{d}(w_{t-\frac{1}{2}}[i] - w^*[i])^2\{\frac{\hat{v}_t^{1/2}[i]}{\eta_t} - \frac{\hat{v}_{t-1}^{1/2}[i]}{\eta_{t-1}}\} \\
&+ \sum_{t=1}^{T}\frac{\eta_t}{(1-\beta_1)}\|\theta_t\|_{\psi_t^*}^2 + \sum_{t=1}^{T}\frac{\beta_1}{2\eta_t(1-\beta_1)}\|w_{t-\frac{1}{2}} - w^*\|_{\psi_t}^2, \\
&\leq \frac{1}{2\eta_1(1-\beta_1)}D_\infty^2\sum_{i=1}^{d}\hat{v}_1[i]^2 + \frac{1}{2(1-\beta_1)}D_\infty^2\sum_{t=2}^{T}\sum_{i=1}^{d}\{\frac{\hat{v}_t^{1/2}[i]}{\eta_t} - \frac{\hat{v}_{t-1}^{1/2}[i]}{\eta_{t-1}}\} + \sum_{t=1}^{T}\frac{\eta_t}{(1-\beta_1)}\|\theta_t\|_{\psi_t^*}^2 \\
&+ \sum_{t=1}^{T}\frac{\beta_1}{2\eta_t(1-\beta_1)}\|w_{t-\frac{1}{2}} - w^*\|_{\psi_t}^2 \\
&= \frac{1}{2\eta_T(1-\beta_1)}D_\infty^2\sum_{i=1}^{d}\hat{v}_T[i]^2 + \sum_{t=1}^{T}\frac{\eta_t}{(1-\beta_1)}\|\theta_t\|_{\psi_t^*}^2 + \sum_{t=1}^{T}\frac{\beta_1}{2\eta_t(1-\beta_1)}\|w_{t-\frac{1}{2}} - w^*\|_{\psi_t}^2 \\
&\leq \frac{1}{2\eta_T(1-\beta_1)}D_\infty^2\sum_{i=1}^{d}\hat{v}_T[i]^2 + \sum_{t=1}^{T}\frac{\eta_t}{(1-\beta_1)}\|\theta_t\|_{\psi_t^*}^2 + D_\infty^2\sum_{t=1}^{T}\sum_{i=1}^{d}\frac{\beta_1\hat{v}_t[i]^{1/2}}{2\eta_t(1-\beta_1)},
\end{aligned}
\tag{14}
$$

where the last equality is by telescoping sum. $\qquad\square$

## C   PROOF OF LEMMA 2

**Lemma 2**  $\sum_{t=1}^{T} \langle w_t - w_{t-\frac{1}{2}}, g_t - h_t \rangle + \sum_{t=1}^{T} \langle w_t - w_{t-\frac{1}{2}}, h_t \rangle \leq \sum_{t=1}^{T} \frac{\eta_t}{2} \|g_t - h_t\|^2_{(\frac{1-\beta_1}{8}\psi_{t-1})^*} - \frac{3}{2\eta_t} \|w_t - w_{t-\frac{1}{2}}\|^2_{\frac{1-\beta_1}{8}\psi_{t-1}}.$

*Proof.* From the update, $w_t = w_{t-\frac{1}{2}} - \eta_t \frac{4}{1-\beta_1} \frac{h_t}{\sqrt{v_{t-1}}}$,

$$
\begin{aligned}
\langle w_t - w_{t-\frac{1}{2}}, h_t \rangle &= -\eta_t \langle h_t, \frac{4}{1-\beta_1} \frac{h_t}{\mathrm{diag}\{v_t\}^{-1/2}} \rangle \\
&= -\frac{2}{\eta_t} \langle \eta_t \frac{4}{1-\beta_1} \frac{h_t}{\mathrm{diag}\{v_t\}^{1/2}}, \frac{1-\beta_1}{8} \mathrm{diag}\{v_t\}^{1/2} \eta_t \frac{4}{1-\beta_1} \frac{h_t}{\mathrm{diag}\{v_t\}^{1/2}} \rangle \\
&= -\frac{2}{\eta_t} \|w_t - w_{t-\frac{1}{2}}\|^2_{\frac{1-\beta_1}{8}\psi_{t-1}}.
\end{aligned}
\tag{15}
$$

Therefore, we have that

$$
\begin{aligned}
\sum_{t=1}^{T} \langle w_t - w_{t-\frac{1}{2}}, g_t - h_t \rangle + \sum_{t=1}^{T} \langle w_t - w_{t-\frac{1}{2}}, h_t \rangle &= \sum_{t=1}^{T} \langle w_t - w_{t-\frac{1}{2}}, g_t - h_t \rangle - \frac{2}{\eta_t} \|w_t - w_{t-\frac{1}{2}}\|^2_{\frac{1-\beta_1}{8}\psi_{t-1}} \\
&\overset{(a)}{\leq} \sum_{t=1}^{T} \|w_t - w_{t-\frac{1}{2}}\|_{\frac{1-\beta_1}{8}\psi_{t-1}} \|g_t - h_t\|_{(\frac{1-\beta_1}{8}\psi_{t-1})^*} - \frac{2}{\eta_t} \|w_t - w_{t-\frac{1}{2}}\|^2_{\frac{1-\beta_1}{8}\psi_{t-1}} \\
&\overset{(b)}{\leq} \sum_{t=1}^{T} \frac{1}{2\eta_t} \|w_t - w_{t-\frac{1}{2}}\|^2_{\frac{1-\beta_1}{8}\psi_{t-1}} + \frac{\eta_t}{2} \|g_t - h_t\|^2_{(\frac{1-\beta_1}{8}\psi_{t-1})^*} - \frac{2}{\eta_t} \|w_t - w_{t-\frac{1}{2}}\|^2_{\frac{1-\beta_1}{8}\psi_{t-1}} \\
&= \sum_{t=1}^{T} \frac{\eta_t}{2} \|g_t - h_t\|^2_{(\frac{1-\beta_1}{8}\psi_{t-1})^*} - \frac{3}{2\eta_t} \|w_t - w_{t-\frac{1}{2}}\|^2_{\frac{1-\beta_1}{8}\psi_{t-1}},
\end{aligned}
\tag{16}
$$

where (a) is by Hölder's inequality and (b) is by Young's inequality.

$\square$

## D   PROOF OF THEOREM 1

**Theorem 1**  *Denote* $\gamma := \beta_1/\sqrt{\beta_2} < 1$ *and* $D_\infty = \max_t \|w_{t-\frac{1}{2}} - w^*\|_\infty$. *Then,*

$$
\begin{aligned}
Regret_T &\leq \frac{1}{2\eta_T(1-\beta_1)} D_\infty^2 \sum_{i=1}^{d} \hat{v}_T[i]^2 + D_\infty^2 \sum_{t=1}^{T} \sum_{i=1}^{d} \frac{\beta_1 \hat{v}_t[i]^{1/2}}{2\eta_t(1-\beta_1)} \\
&\quad + \sum_{t=1}^{T} 2\eta_t(1-\beta_1)\|g_t - m_t\|^2_{\psi_t^*} + \sum_{t=1}^{T} \frac{\eta_t}{2} \|g_t - h_t\|^2_{(\frac{1-\beta_1}{8}\psi_{t-1})^*}.
\end{aligned}
\tag{17}
$$

*Proof.* Recall that from (7), we show that

$$
\begin{aligned}
\mathrm{Regret}_T &\leq \frac{1}{2\eta_T(1-\beta_1)} D_\infty^2 \sum_{i=1}^{d} \hat{v}_T[i]^2 + \sum_{t=1}^{T} \frac{\eta_t}{(1-\beta_1)} \|\theta_t\|^2_{\psi_t^*} \\
&\quad + D_\infty^2 \sum_{t=1}^{T} \sum_{i=1}^{d} \frac{\beta_1 \hat{v}_t[i]^{1/2}}{2\eta_t(1-\beta_1)} + \sum_{t=1}^{T} \frac{\eta_t}{2} \|g_t - h_t\|^2_{(\frac{1-\beta_1}{8}\psi_{t-1})^*} - \frac{3}{2\eta_t} \|w_t - w_{t-\frac{1}{2}}\|^2_{\frac{1-\beta_1}{8}\psi_{t-1}}.
\end{aligned}
\tag{18}
$$

To proceed, let us analyze $\sum_{t=1}^{T} \frac{\eta_t}{(1-\beta_1)} \|\theta_t\|^2_{\psi_t^*} - \frac{3}{2\eta_t} \|w_t - w_{t-\frac{1}{2}}\|^2_{\frac{1-\beta_1}{8}\psi_{t-1}}$ above. Notice that

$$
\frac{\eta_t}{(1-\beta_1)} \|\theta_t\|^2_{\psi_t^*} = \frac{\eta_t}{(1-\beta_1)} \sum_{i=1}^{d} \frac{\theta_t[i]^2}{\sqrt{\hat{v}_t[i]}},
\tag{19}
$$

and

$$
\frac{3}{2\eta_t} \|w_t - w_{t-\frac{1}{2}}\|^2_{\frac{1-\beta_1}{8}\psi_{t-1}} = \frac{3}{2\eta_t} \langle \eta_t \frac{4}{1-\beta_1} \frac{h_t}{\sqrt{\hat{v}_{t-1}}}, \frac{1-\beta_1}{8} \mathrm{diag}(\hat{v}_{t-1})^{1/2} \eta_t \frac{4}{1-\beta_1} \frac{h_t}{\sqrt{\hat{v}_{t-1}}} \rangle
$$
$$
= \frac{3\eta_t}{1-\beta_1} \sum_{i=1}^{d} \frac{h_t[i]^2}{\sqrt{\hat{v}_{t-1}[i]}},
$$

(20)

where we use the update rule, $w_t = w_{t-\frac{1}{2}} - \eta_t \frac{4}{1-\beta_1} \frac{h_t}{\sqrt{\hat{v}_{t-1}}}$. Therefore, we have that

$$
\sum_{t=1}^{T} \frac{\eta_t}{(1-\beta_1)} \|\theta_t\|^2_{\psi_t^*} - \frac{3}{2\eta_t} \|w_t - w_{t-\frac{1}{2}}\|^2_{\frac{1-\beta_1}{2}\psi_{t-1}}
$$
$$
\leq \sum_{t=1}^{T} \frac{\eta_t}{(1-\beta_1)} \sum_{i=1}^{d} \frac{\theta_t[i]^2}{\sqrt{\hat{v}_t[i]}} - \sum_{t=1}^{T} \frac{3\eta_t}{(1-\beta_1)} \sum_{i=1}^{d} \frac{h_t[i]^2}{\sqrt{\hat{v}_{t-1}[i]}}
$$
$$
\leq \sum_{t=1}^{T} \frac{\eta_t}{(1-\beta_1)} \sum_{i=1}^{d} \frac{\theta_t[i]^2}{\sqrt{\hat{v}_t[i]}} - \sum_{t=1}^{T} \frac{\eta_t}{(1-\beta_1)} \sum_{i=1}^{d} \frac{h_t[i]^2}{\sqrt{\hat{v}_t[i]}},
$$
$$
= \sum_{t=1}^{T} \frac{\eta_t}{(1-\beta_1)} \sum_{i=1}^{d} \frac{(\theta_t[i]^2 - h_t[i]^2)}{\sqrt{\hat{v}_t[i]}}
$$

(21)

$$
\leq \sum_{t=1}^{T} \frac{2\eta_t}{(1-\beta_1)} \sum_{i=1}^{d} \frac{(\theta_t[i] - h_t[i])^2}{\sqrt{\hat{v}_t[i]}}
$$
$$
= \sum_{t=1}^{T} \frac{2\eta_t}{(1-\beta_1)} \|\theta_t - h_t\|^2_{\psi_t^*},
$$
$$
= \sum_{t=1}^{T} 2\eta_t (1-\beta_1) \|g_t - m_t\|^2_{\psi_t^*},
$$

where the second inequality is due to that the sequence $\{\hat{v}_t[i]\}$ is non-decreasing and the last equality is because $\theta_t - h_t = \beta_1\theta_{t-1} + (1-\beta_1)(g_t) - (\beta_1\theta_{t-1} + (1-\beta_1)(m_t)) = (1-\beta_1)(g_t - m_t)$.

To summarize,

$$
\mathrm{Regret}_T \leq \frac{1}{2\eta_T(1-\beta_1)} D_\infty^2 \sum_{i=1}^{d} \hat{v}_T[i]^2 + D_\infty^2 \sum_{t=1}^{T} \sum_{i=1}^{d} \frac{\beta_1 \hat{v}_t[i]^{1/2}}{2\eta_t(1-\beta_1)}
$$
$$
+ \sum_{t=1}^{T} 2\eta_t(1-\beta_1)\|g_t - m_t\|^2_{\psi_t^*} + \sum_{t=1}^{T} \frac{\eta_t}{2} \|g_t - h_t\|^2_{(\frac{1-\beta_1}{8}\psi_{t-1})^*}.
$$

(22)

$\square$

# E   PROOF OF THEOREM 2

**Theorem 2** *There exists a convex online learning problem such that* ADAM-DISZ *has nonzero average regret (i.e.* $\frac{Regret}{T} \neq o(T)$)

We basically follow the same setting as Theorem 1 of Reddi et al. (2018). In each round, the loss function $\ell_t(w)$ is linear and the learner's decision space is $\mathcal{K} = [-1, 1]$

$$f_t(w) = \begin{cases} Cw \text{ , for } t \bmod 3 = 1 \\ -w \text{ , otherwise} \end{cases} \quad ,$$

where $C \geq 4$. For this loss function sequences, the point $w = -1$ achieves the minimum regret, *i.e.* $-1 = \arg\min_{w \in \mathcal{K}} \sum_{t=1}^{T} f_t(w)$ when $T \to \infty$.

Consider the execution of ADAM-DISZ with $\beta_1 = 0$, $\beta_2 = \frac{1}{1+2C^2}$, $\eta_t = \frac{\eta}{\sqrt{t}}$ with $\eta < \frac{1}{3}\sqrt{1 - \beta_2}$. Notice that $\beta_1^2/\sqrt{\beta_2} < 1$ in this case, which satisfies the conditions of Kingma & Ba (2015).

The goal is to show that for all $t$, $w_{3t+1} = 1$, and $w_t > 0$. Let us denote $\hat{w}_{t+1} := w_t - 2\eta_t \frac{\theta_t}{\sqrt{v_t}} + \eta_t \frac{\theta_{t-1}}{\sqrt{v_{t-1}}}$ and $w_{t+1} := \Pi_k[\hat{w}_{t+1}]$. Assume the initial point is 1. As pointed it out by Reddi et al. (2018), the assumption of the initial point is without loss of generality. If the initial point is not 1, then one can translate the coordinate system to a new coordinate system so that the initial point is 1 and then choose the loss function sequences for the new coordinate system. Therefore, the base case $x_1 = 1$ is true.

Now assume that for some $t > 0$, $x_{3t+1} = 1$ and $x_s > 0$ for all $s \leq 3t + 1$. Observe that.

$$\nabla f_t(w) = \begin{cases} C \text{ , for } t \bmod 3 = 1 \\ -1 \text{ , otherwise} \end{cases} \quad ,$$

According to the update of ADAM-DISZ

$$\begin{aligned}
\hat{w}_{3t+2} &= w_{3t+1} - \frac{2\eta C}{\sqrt{(3t+1)(\beta_2 v_{3t} + (1-\beta_2)C^2)}} - \frac{\eta}{\sqrt{(3t+1)(\beta_2 v_{3t-1} + (1-\beta_2))}} \\
&= 1 - \frac{2\eta C}{\sqrt{(3t+1)(\beta_2 v_{3t} + (1-\beta_2)C^2)}} - \frac{\eta}{\sqrt{(3t+1)(\beta_2 v_{3t-1} + (1-\beta_2))}} \\
&\geq 1 - \frac{2\eta}{\sqrt{(3t+1)(1-\beta_2)}} - \frac{\eta}{\sqrt{(3t+1)(1-\beta_2)}} \\
&\geq 1 - \frac{3\eta}{\sqrt{(3t+1)(1-\beta_2)}} > 0.
\end{aligned} \tag{23}$$

So, $\hat{w}_{3t+2} = w_{3t+2}$.

$$\hat{w}_{3t+3} = w_{3t+2} + \frac{2\eta}{\sqrt{(3t+2)(\beta_2 v_{3t+1} + (1-\beta_2))}} + \frac{C\eta}{\sqrt{(3t+2)(\beta_2 v_{3t} + (1-\beta_2)C^2)}} > 0. \tag{24}$$

For $\hat{w}_{3t+4}$, let us first consider the case that $\hat{w}_{3t+3} < 1$

$$
\hat{w}_{3t+4} = \min(\hat{w}_{3t+3}, 1) + \frac{2\eta}{\sqrt{(3t+3)(\beta_2 v_{3t+2} + (1 - \beta_2))}} - \frac{\eta}{\sqrt{(3t+3)(\beta_2 v_{3t+1} + (1 - \beta_2))}}
$$

$$
= w_{3t+2} + \frac{2\eta}{\sqrt{(3t+2)(\beta_2 v_{3t+1} + (1 - \beta_2))}} + \frac{C\eta}{\sqrt{(3t+2)(\beta_2 v_{3t} + (1 - \beta_2)C^2)}}
$$

$$
+ \frac{2\eta}{\sqrt{(3t+3)(\beta_2 v_{3t+2} + (1 - \beta_2))}} - \frac{\eta}{\sqrt{(3t+3)(\beta_2 v_{3t+1} + (1 - \beta_2))}}
$$

$$
\geq 1 - \frac{2\eta C}{\sqrt{(3t+1)(\beta_2 v_{3t} + (1 - \beta_2)C^2)}} - \frac{\eta}{\sqrt{(3t+1)(\beta_2 v_{3t-1} + (1 - \beta_2))}}
$$

$$
+ \frac{\eta}{\sqrt{(3t+2)(\beta_2 v_{3t+1} + (1 - \beta_2))}} + \frac{C\eta}{\sqrt{(3t+2)(\beta_2 v_{3t} + (1 - \beta_2)C^2)}} + \frac{2\eta}{\sqrt{(3t+3)(\beta_2 v_{3t+2} + (1 - \beta_2))}}
$$

$$
= 1 - (1 - \frac{\sqrt{3t+1}}{\sqrt{3t+2}}) \frac{\eta C}{\sqrt{(3t+1)(\beta_2 v_{3t} + (1 - \beta_2)C^2)}}
$$

$$
- \frac{\eta}{\sqrt{(3t+1)(\beta_2 v_{3t-1} + (1 - \beta_2))}} + \frac{\eta}{\sqrt{(3t+1)(\beta_2 v_{3t+1} + (1 - \beta_2))}}
$$

$$
- \frac{\eta C}{\sqrt{(3t+1)(\beta_2 v_{3t} + (1 - \beta_2)C^2)}} + \frac{2\eta}{\sqrt{(3t+3)(\beta_2 v_{3t+2} + (1 - \beta_2))}}
$$

$$
\geq 1 - (1 - \frac{\sqrt{3t+1}}{\sqrt{3t+2}}) \frac{\eta}{\sqrt{(3t+1)(1 - \beta_2)}}
$$

$$
- \frac{2\eta}{\sqrt{(3t+1)(1 - \beta_2)}} + \frac{\eta}{\sqrt{(3t+1)(\beta_2 C^2 + (1 - \beta_2))}} + \frac{2\eta}{\sqrt{(3t+3)(\beta_2 C^2 + (1 - \beta_2))}}
$$

$$
\overset{(a)}{=} 1 - (1 - \frac{\sqrt{3t+1}}{\sqrt{3t+2}}) \frac{\eta}{\sqrt{(3t+1)(1 - \beta_2)}}
$$

$$
- \frac{(2 - \frac{1}{\sqrt{3/2}})\eta}{\sqrt{(3t+1)(1 - \beta_2)}} + \frac{2\eta}{\sqrt{(3t+3)(\beta_2 C^2 + (1 - \beta_2))}}
$$

$$
\geq 1 - (1 - \frac{\sqrt{3t+1}}{\sqrt{3t+2}}) \frac{\eta}{\sqrt{(3t+1)(1 - \beta_2)}}
$$

$$
- \frac{(2 - \frac{1}{\sqrt{3/2}})\eta}{\sqrt{(3t+1)(1 - \beta_2)}} + \frac{\sqrt{3}\eta}{\sqrt{(3t+1)(\beta_2 C^2 + (1 - \beta_2))}}
$$

$$
\overset{(a)}{=} 1 - (1 - \frac{\sqrt{3t+1}}{\sqrt{3t+2}}) \frac{\eta}{\sqrt{(3t+1)(1 - \beta_2)}} - \frac{(2 - \frac{1}{\sqrt{3/2}})\eta}{\sqrt{(3t+1)(1 - \beta_2)}} + \frac{\sqrt{2}\eta}{\sqrt{(3t+1)(1 - \beta_2)}}
$$

$$
\geq 1 - (1 - \frac{\sqrt{3t+1}}{\sqrt{3t+2}}) \frac{\eta}{\sqrt{(3t+1)(1 - \beta_2)}} + 0.23 \frac{\eta}{\sqrt{(3t+1)(1 - \beta_2)}}
$$

$$
\overset{(b)}{\geq} 1 - 0.12 \frac{\eta}{\sqrt{(3t+1)(1 - \beta_2))}} + 0.23 \frac{\eta}{\sqrt{(3t+1)(1 - \beta_2)}} \geq 1
$$

(25)

where (a) uses $\sqrt{\frac{\beta_2 C^2 + (1 - \beta_2)}{3/2}} = \sqrt{1 - \beta_2}$ for the choice of $\beta_2 = 1/(1 + 2C^2)$, (b) is because $1 - \frac{\sqrt{3t+1}}{\sqrt{3t+2}}$ is a decreasing function of $t$ and has its largest value at $t = 1$ for any $t > 0$, which is $1 - (\frac{\sqrt{4}}{\sqrt{5}}) \leq 0.12$. To summarize, we have that $w_{3t+4} = 1$.

Now if it is the case that $\hat{w}_{3t+3} \geq 1$

$$
\hat{w}_{3t+4} = \min(\hat{w}_{3t+3}, 1) + \frac{2\eta}{\sqrt{(3t+3)(\beta_2 v_{3t+2} + (1 - \beta_2))}} - \frac{\eta}{\sqrt{(3t+3)(\beta_2 v_{3t+1} + (1 - \beta_2))}}
$$

$$
= 1 + \frac{2\eta}{\sqrt{(3t+3)(\beta_2 v_{3t+2} + (1 - \beta_2))}} - \frac{\eta}{\sqrt{(3t+3)(\beta_2 v_{3t+1} + (1 - \beta_2))}} > 1
$$

(26)

where the last inequality is because

$$2\sqrt{\beta_2 v_{3t+1} + (1 - \beta_2)} \geq \sqrt{\beta_2 v_{3t+2} + (1 - \beta_2)} \qquad (27)$$

To see this,

$$\sqrt{\beta_2 v_{3t+2} + (1 - \beta_2)} = \sqrt{\beta_2(\beta_2 v_{3t+1} + (1 - \beta_2)) + (1 - \beta_2)} = \sqrt{(\beta_2^2) v_{3t+1} + 1 - \beta_2^2} \qquad (28)$$

So, the above inequality is equivalent to

$$2\sqrt{\beta_2 v_{3t+1} + (1 - \beta_2)} \geq \sqrt{(\beta_2^2) v_{3t+1} + 1 - \beta_2^2}, \qquad (29)$$

which is true. This means that $w_{3t+4} = 1$. Therefore, we have completed the induction.

To summarize, we have $\ell_{3t+1}(w_{3t+1}) + \ell_{3t+2}(w_{3t+2}) + \ell_{3t+3}(w_{3t+3}) - \ell_{3t+1}(-1) - \ell_{3t+2}(w_{-1}) - \ell_{3t+3}(-1) \geq 2C - 4$. That is, for every 3 steps, the algorithms suffers regret at least $2C - 4$, this means that the regret over $T$ rounds would be $(2C - 4)T/3$, which is not sublinear to $T$. Now we have completed the proof. One may follow the analysis of Theorem 2 of Reddi et al. (2018) to generalize the result so that ADAM-DISZ does not converge for any $\beta_1, \beta_2 \in [0, 1)$ such that $\beta_1 < \sqrt{\beta_2}$.

## F  OPTIMISTIC-ADAGRAD

The optimistic update step can also be extended to other algorithms as well. For example, based on ADAGRAD, we propose OPTIMISTIC-ADAGRAD by including the optimistic update step.

---

**Algorithm 5** OPTIMISTIC-ADAGRAD (UNCONSTRAINED)

---

1: Required: parameter $\eta$.
2: Init: $w_1$.
3: **for** $t = 1$ to $T$ **do**
4:    Get current gradient $g_t$ at $w_t$.
5:    $w_{t+\frac{1}{2}} = w_{t-\frac{1}{2}} - \eta \text{diag}(G_t)^{-1/2} g_t$

  where $\text{diag}(G_t) := \begin{pmatrix} \sum_{s=1}^t g_t[1]^2 & 0 & 0 \\ 0 & \sum_{s=1}^t g_t[2]^2 & 0 \\ 0 & 0 & . \end{pmatrix}$

6:    $w_{t+1} = w_{t+\frac{1}{2}} - \eta \text{diag}(G_t)^{-1/2} m_{t+1}$,
     where $m_{t+1}$ is the guess of $g_{t+1}$.
7: **end for**

---

### F.1  ANALYSIS:

We provide the regret analysis here. Let us recall the notations and assumptions first. We denote the Mahalanobis norm $\|\cdot\|_H = \sqrt{\langle\cdot, H\cdot\rangle}$ for some PSD matrix $H$. We let $\psi_t(x) := \langle x, \text{diag}\{\hat{v}_t\}^{1/2} x\rangle$ for the PSD matrix $\text{diag}\{\hat{v}_t\}$, where $\text{diag}\{\hat{v}_t\}$ represents the diagonal matrix such that its $i_{th}$ diagonal element is $\hat{v}_t[i]$ in Algorithm 2. Consequently, $\psi_t(\cdot)$ is 1-strongly convex with respect to the norm $\|\cdot\|_{\psi_t} := \sqrt{\langle\cdot, \text{diag}\{\hat{v}_t\}^{1/2}\cdot\rangle}$. Namely, $\psi_t(\cdot)$ satisfies $\psi_t(u) \geq \psi_t(v) + \langle\psi_t(v), u - v\rangle + \frac{1}{2}\|u - v\|_{\psi_t}^2$ for any point $u, v$. A consequence of 1-strongly convexity of $\psi_t(\cdot)$ is that $B_{\psi_t}(u, v) \geq \frac{1}{2}\|u-v\|_{\psi_t}^2$, where the Bregman divergence $B_{\psi_t}(u, v)$ is defined as $B_{\psi_t}(u, v) := \psi_t(u) - \psi_t(v) - \langle\psi_t(v), u - v\rangle$ and $\psi_t$ is called the distance generating function. We can also define the associate dual norm as $\psi_t^*(x) := \langle x, \text{diag}\{\hat{v}_t\}^{-1/2} x\rangle$. We assume that the model parameter $w$ is in $d$-dimensional space. That is, $w \in \mathbb{R}^d$. It suffices to analyze Algorithm 6, which holds for any convex set $\mathcal{K}$. The algorithm reduces to Algorithm 5 when $\mathcal{K} = \mathbb{R}^d$.

---

**Algorithm 6** OPTIMISTIC-ADAGRAD

1: Required: parameter $\eta_t$.
2: Init: $w_1$.
3: **for** $t = 1$ to $T$ **do**
4:    Get current gradient $g_t$ at $w_t$.
5:    $w_{t+\frac{1}{2}} = \arg\min_{w \in \mathcal{K}} \eta_t \langle w, g_t \rangle + B_{\psi_t}(w, w_{t-\frac{1}{2}})$.
6:    $w_{t+1} = \arg\min_{w \in \mathcal{K}} \eta_{t+1} \langle w, m_{t+1} \rangle + B_{\psi_t}(w, w_{t+\frac{1}{2}})$
      where $m_{t+1}$ is the guess of $g_{t+1}$.
7: **end for**

---

By regret decomposition, we have that

$$
\text{Regret}_T := \sum_{t=1}^{T} \ell_t(w_t) - \min_{w \in \mathcal{K}} \sum_{t=1}^{T} \ell_t(w) \leq \sum_{t=1}^{T} \langle w_t - w^*, \nabla \ell_t(w_t) \rangle
$$

$$
= \sum_{t=1}^{T} \langle w_t - w_{t+\frac{1}{2}}, \nabla \ell_t(w_t) - m_t \rangle + \langle w_t - w_{t+\frac{1}{2}}, m_t \rangle + \langle w_{t+\frac{1}{2}} - w^*, \nabla \ell_t(w_t) \rangle.
$$
(30)

Now we are going to exploit a useful inequality: for any update of the form $\hat{w} = \arg\min_{w \in \mathcal{K}} \langle w, \theta \rangle + B_\psi(w, v)$, we have that

$$
\langle \hat{w} - u, \theta \rangle \leq B_\psi(u, v) - B_\psi(u, \hat{w}) - B_\psi(\hat{w}, v),
$$
(31)

for any $u \in \mathcal{K}$. Using the above inequality, we have

$$
\langle w_t - w_{t+\frac{1}{2}}, m_t \rangle \leq \frac{1}{\eta_t} (B_{\psi_{t-1}}(w_{t+\frac{1}{2}}, w_{t-\frac{1}{2}}) - B_{\psi_{t-1}}(w_{t+\frac{1}{2}}, w_t) - B_{\psi_{t-1}}(w_t, w_{t-\frac{1}{2}})),
$$
(32)

and

$$
\langle w_{t+\frac{1}{2}} - w^*, \nabla \ell_t(w_t) \rangle \leq \frac{1}{\eta_t} (B_{\psi_t}(w^*, w_{t-\frac{1}{2}}) - B_{\psi_t}(w^*, w_{t+\frac{1}{2}}) - B_{\psi_t}(w_{t+\frac{1}{2}}, w_{t-\frac{1}{2}})).
$$
(33)

So, by (30) (32) and (33), we obtain

$$
\text{Regret}_T \leq \sum_{t=1}^{T} \langle w_t - w_{t+\frac{1}{2}}, \nabla \ell_t(w_t) - m_t \rangle + \langle w_t - w_{t+\frac{1}{2}}, m_t \rangle + \langle w_{t+\frac{1}{2}} - w^*, \nabla \ell_t(w_t) \rangle
$$

$$
\leq \sum_{t=1}^{T} \| w_t - w_{t+\frac{1}{2}} \|_{\psi_{t-1}} \| \nabla f(w_t) - m_t \|_{\psi_{t-1}^*}
$$

$$
+ \frac{1}{\eta_t} \big( B_{\psi_{t-1}}(w_{t+\frac{1}{2}}, w_{t-\frac{1}{2}}) - B_{\psi_{t-1}}(w_{t+\frac{1}{2}}, w_t) - B_{\psi_{t-1}}(w_t, w_{t-\frac{1}{2}})
$$

$$
+ B_{\psi_t}(w^*, w_{t-\frac{1}{2}}) - B_{\psi_t}(w^*, w_{t+\frac{1}{2}}) - B_{\psi_t}(w_{t+\frac{1}{2}}, w_{t-\frac{1}{2}}) \big)
$$

$$
\leq \sum_{t=1}^{T} \frac{1}{2\eta_t} \| w_t - w_{t+\frac{1}{2}} \|_{\psi_{t-1}}^2 + \frac{\eta_t}{2} \| \nabla f(w_t) - m_t \|_{\psi_{t-1}^*}^2
$$

$$
+ \frac{1}{\eta_t} \big( B_{\psi_{t-1}}(w_{t+\frac{1}{2}}, w_{t-\frac{1}{2}}) - \frac{1}{2} \| w_{t+\frac{1}{2}} - w_t \|_{\psi_{t-1}}^2 - B_{\psi_{t-1}}(w_t, w_{t-\frac{1}{2}})
$$

$$
+ B_{\psi_t}(w^*, w_{t-\frac{1}{2}}) - B_{\psi_t}(w^*, w_{t+\frac{1}{2}}) - B_{\psi_t}(w_{t+\frac{1}{2}}, w_{t-\frac{1}{2}}) \big)
$$

$$
\leq \sum_{t=1}^{T} \frac{\eta_t}{2} \| \nabla f(w_t) - m_t \|_{\psi_{t-1}^*}
$$

$$
+ \frac{1}{\eta_t} \big( B_{\psi_t}(w^*, w_{t-\frac{1}{2}}) - B_{\psi_t}(w^*, w_{t+\frac{1}{2}}) + B_{\psi_{t-1}}(w_{t+\frac{1}{2}}, w_{t-\frac{1}{2}}) - B_{\psi_t}(w_{t+\frac{1}{2}}, w_{t-\frac{1}{2}}) \big),
$$
(34)

where the third inequality is because

$$
\| w_t - w_{t+\frac{1}{2}} \|_{\psi_{t-1}} \| \nabla f(w_t) - m_t \|_{\psi_{t-1}^*} = \inf_{\beta > 0} \frac{1}{2\beta} \| w_t - w_{t+\frac{1}{2}} \|_{\psi_{t-1}}^2 + \frac{\beta}{2} \| \nabla f(w_t) - m_t \|_{\psi_{t-1}^*}^2,
$$
(35)

and that $\psi_{t-1}(\cdot)$ is 1-strongly convex with respect to $\|\cdot\|_{\psi_{t-1}}$. To proceed, notice that, by definition of Bregman divergence, we also have

$$
\begin{aligned}
B_{\psi_{t+1}}(w^*, w_{t+\frac{1}{2}}) - B_{\psi_t}(w^*, w_{t+\frac{1}{2}}) &= \langle w^* - w_{t+\frac{1}{2}}, \mathrm{diag}(s_{t+1} - s_t)(w^* - w_{t+\frac{1}{2}}) \rangle \\
&\leq \max_i (w^*[i] - w_{t+\frac{1}{2}}[i])^2 \|s_{t+1} - s_t\|_1.
\end{aligned}
\tag{36}
$$

and

$$
B_{\psi_{t-1}}(w_{t+\frac{1}{2}}, w_{t-\frac{1}{2}}) - B_{\psi_t}(w_{t+\frac{1}{2}}, w_{t-\frac{1}{2}}) = \langle w_{t+\frac{1}{2}} - w_{t-\frac{1}{2}}, \mathrm{diag}(s_{t-1} - s_t)(w_{t+\frac{1}{2}} - w_{t-\frac{1}{2}}) \rangle \leq 0.
\tag{37}
$$

where the vector $s_t \in \mathbb{R}^d$ is defined as follows: its $i_{th}$ entry $s_t[i]$ is

$$
s_t[i] := \sqrt{\sum_{s=1}^t g_s[i]^2}.
\tag{38}
$$

Now we have

$$
\begin{aligned}
\mathrm{Regret}_T &\leq \sum_{t=1}^T \frac{\eta_t}{2} \|\nabla f(w_t) - m_t\|_{\psi_{t-1}^*}^2 \\
&\quad + \frac{1}{\eta_t} \Big( B_{\psi_t}(w^*, w_{t-\frac{1}{2}}) - B_{\psi_t}(w^*, w_{t+\frac{1}{2}}) + B_{\psi_{t-1}}(w_{t+\frac{1}{2}}, w_{t-\frac{1}{2}}) - B_{\psi_t}(w_{t+\frac{1}{2}}, w_{t-\frac{1}{2}}) \Big) \\
&\leq \frac{1}{\eta_1} B_{\psi_1}(w^*, w_{1/2}) + \sum_{t=1}^T \frac{\eta_t}{2} \|\nabla f(w_t) - m_t\|_{\psi_{t-1}^*}^2 + \sum_{t=1}^{T-1} \frac{1}{\eta_t} \max_i (w^*[i] - w_{t+\frac{1}{2}}[i])^2 \|s_{t+1} - s_t\|_1 \\
&\leq \frac{1}{\eta_1} B_{\psi_1}(w^*, w_{1/2}) + \frac{1}{\eta_{\min}} \max_{t \leq T} \|w^* - w_{t-\frac{1}{2}}\|_\infty^2 \sum_{i=1}^d \|g_{1:T}[i]\|_2 + \sum_{t=1}^T \frac{\eta_t}{2} \|\nabla f(w_t) - m_t\|_{\psi_{t-1}^*}^2.
\end{aligned}
\tag{39}
$$

Therefore, we conclude the following theorem.

**Theorem 3.** *For any learning rate $\eta_t := \eta$,* OPTIMISTIC-ADAGRAD *(Algorithm 6) has regret*

$$
Regret_T \leq \frac{1}{\eta} B_{\psi_1}(w^*, w_{1/2}) + \frac{1}{\eta} \max_{t \leq T} \|w^* - w_{t-\frac{1}{2}}\|_\infty^2 \sum_{i=1}^d \|g_{1:T}[i]\|_2 + \sum_{t=1}^T \frac{\eta}{2} \|\nabla f(w_t) - m_t\|_{\psi_{t-1}^*}^2
\tag{40}
$$

One need to compare it with the bound of ADAGRAD Duchi et al. (2011), which has regret [3]

$$
\mathrm{Regret}_T = O(\max_{t \leq T} \|w^* - w_{t-\frac{1}{2}}\|_\infty^2 \sum_{i=1}^d \|g_{1:T}[i]\|_2).
\tag{41}
$$

We see that when $m_t$ predicts $\nabla f(w_t)$ very well, the last term is dominated by the second one. One can set $\eta$ of OPTIMISTIC-ADAGRAD to be a large step size so that OPTIMISTIC-ADAGRAD has a smaller regret than ADAGRAD.

---

[3]The bound here is the result of optimizing over $\eta$ of ADAGRAD.

