# OpenReview forum: "Optimistic Acceleration for Optimization"
_ICLR.cc/2019/Conference_

### Official Review · AnonReviewer3 · 2018-10-28
**Main idea is not sufficiently novel and technical and empirical results are not convincing enough**

**Rating:** 4
**Confidence:** 4

**Review:**

This paper combines recent results in online learning and convex optimization, specifically adaptivity, momentum, and optimism. The authors add an optimistic gradient prediction step into the AMSGrad algorithm proposed by Reddi et al, 2018. Moreover, they propose using the RMPE algorithm of Scieur et al, 2016 to come up with the gradient prediction step. The new method that they introduce is called Optimistic AMSGrad, and the authors present both theoretical guarantees as well as numerical experiments justifying this new method.

The paper is relatively well-written, and the authors do a good job of explaining recent work on adaptivity, momentum, and optimism in online learning and convex optimization to motivate their algorithm. The algorithm is also presented clearly, and the fact that the method is accompanied by both a regret bound as well as numerical experiments is appreciated.

At the same time, I found the presentation of this work to be a little misleading. The idea of applying optimism to Adam was already presented in Daskalakis et al, 2018. The algorithm in that paper is, in fact, called "Optimistic Adam". I found it very strange that the authors chose to rename that algorithm in this paper. There are two main differences between Optimistic Adam in Daskalakis et al, 2018 and Optimistic AMSGrad. The first is the extension from Adam to AMSGrad, which involves an extra maximization step (line 7 in Algorithm 2) that is immediate. The second is the choice of gradient prediction method. Since Daskalakis et al, 2018 were concerned with equilibrium convergence, they opted to use the most recent gradient as the prediction. On the other hand, the authors in this work are concerned with general online optimization, so they use a linear combination of past gradients as the prediction, based on a method introduced by Scieur et al, 2016. On its own, I do not find this extensions to be sufficiently novel or significant to merit publication.

The fact that this paper includes theoretical guarantees for Optimistic AMSGrad that were missing in Daskalakis et al, 2018 for Optimistic Adam does make it a little more compelling. However, I found the bound in Theorem 1 to be a little strange in that
(1) it doesn't reduce to the AMSGrad bound when the gradient predictions are 0 and (2) it doesn't seem better than the AMSGrad or optimistic FTRL bounds. The authors claim to justify (2) by saying that the extra g_t - h_t term is O(\sqrt{T}), but the whole appeal of adaptive algorithms is that the \sqrt{T} terms are data-dependent. The empirical results also do not include error bars, which makes it hard to judge their significance.

There were also many grammatical errors and typos in the paper.

Other comments and questions:
1) Page 1: "Their theoretical analysis are the regret analysis in online learning." Grammatical error.
2) Page 2: "The concern is that how to get good m_t". Grammatical error.
3) Page 3: "as the future work". Grammatical error.
4) Page 3: "Nestrov's method". Typo.
5) Page 4: "with input consists of". Grammatical error
6) Page 4: "Call Algorithm 3 with..." What is the computational cost of this step? One of the main benefits of algorithms like AMSGrad is that they run in O(d) time with very mild constants.
7) Page 4: "For this extrapolation method to well well..., the gradient vectors at a specific time span is assumed to be captured by (5). If the gradient does not change significantly, this will be a mild condition." If the gradient doesn't change significantly, then choosing m_t = g_t would also work well, wouldn't it? Can you come up with examples of objectives for which this method makes sense? Even toy ones would strengthen this paper.
8) Page 5: Equation (8). As discussed above, this bound doesn't appear to reduce to the AMSGrad bound for m_t = 0, which makes it a little unsatisfying. The fact that there is an extra expression that isn't in terms of the "gradient prediction error" that one has for optimistic FTRL also makes the bound a little strange.
9) Page 7: "The conduct Optimistic-AMSGrad with different values of r and observe similar performance". You should mention that you show the performance for some of these different values in the appendix.
10) Page 7: "multi-classification problems". Typo.
11) Page 7: Figure 1. Without error bars, it's impossible to tell whether these results are meaningful. Moreover, it's strange to evaluate algorithms with online convex optimization guarantees on off-line non-convex problems.
12) Page 7: "widely studied and playing". Grammatical error.
13) Page 8: "A potential directions". Grammatical error.

---

> ### Author Response · Authors · 2018-11-27
> **Response to AnonReviewer3**
>
> Thank you for the valuable comments and identifying the typos. We have fixed them. Please find as follows our response.
> We've adjusted the names of algorithms and updated a new version accordingly.
>
> == Regarding to Theorem 1:
> The proposed algorithm does not reduce to AMSGrad when $m_{t}=0$. So, the regret bound is different.
> If one remove line 9, or set $h_{t+1}=0$, then the last two terms of the regret bound would disappear,
> which becomes the bound of AMSGrad (namely, (9)).
>
> As AnonReviewer 4 points out, we should also compare the sum of the last two terms in Theorem 1 (namely, (8)) with the last term on (9). We are not claiming that the regret bound of Theorem 1 is always better than that of AMSGrad (namely, (9)).
> Our original discussion actually means that if $m_{t}$ and $g_{t}$ is close, then the last term of (8) would dominate. We then try to upper-bound it in a very loose sense (i.e. (10)). We treat each $\frac{(g_t[i]^2- h_t[i]^2)}{\sqrt{\hat{v}_{t-1}[i]}}$ as a constant and get a $O(\sqrt{T})$ bound. However, in practice, $\sqrt{\hat{v}_{t-1}[i]}$ might grow over time,
> and the growth rate is different for different dimension $i$. If $\sqrt{\hat{v}_{t-1}[i]}$ grows like $O(\sqrt{t})$ then the last term of (8) is just $O(\log T)$, which might be better than that of the last term on (9). One can also get a data dependent bound like the last term on (9). We just wanted to say that the regret bound might be better than that of AMSGrad under certain conditions.

---

> > ### Comment · AnonReviewer3 · 2018-11-28
> > **Response to author response**
> >
> > Thanks for the response.
> >
> > I am aware that the algorithm doesn't reduce to AMSGrad when m_t=0, and I think that this makes it more important to carefully highlight when the new algorithm will outperform existing methods (such as AMSGrad). It's probably not reasonable to do this theoretically on real-world problems (although it would be interesting to show how the terms in question behave in your experiments), but even demonstrating cases where the proposed algorithm is better on toy objectives would be helpful for the reader.

---

### Official Review · AnonReviewer1 · 2018-11-03
**reasonable algorithms, no surprises**

**Rating:** 5
**Confidence:** 4

**Review:**

The paper proposes new online optimization algorithms by adding the idea of optimistic updates to the already popular components of adaptive preconditioning and momentum (as used in AMSGrad and ADAM). Such optimistic schemes attempt to guess the yet-unseen gradients before each update, which can lead to better regret guarantees when the guesses are accurate in a certain sense. This in turn can lead to faster convergence when the resulting algorithm is used in an optimization framework. The specific contribution of the present paper is proving formally that optimistic updates can indeed be combined with advanced methods like ADAM and AMSGrad, also providing a regret analysis of the former algorithm. On the practical front, the authors also propose a method closely resembling Anderson acceleration for guessing the next gradient, and the eventual scheme is shown to work well empirically in training deep neural networks.

The idea of optimistic updates has been popular in recent years within the online-learning literature, and has been used with particularly great success for achieving improved convergence guarantees for learning equilibria in games. More recently, optimistic updates have also appeared in more "practical" settings such as training GANs, where they were shown to improve stability of training. The present paper argues that the idea of optimism can be useful for large-scale optimization as well, if the gradient guesses are chosen appropriately.

I have lukewarm feelings about the paper. On the positive side, the proposed method is a natural and sensible combination of solid technical ideas, and its theoretical analysis appears to be correct. As the authors point out, their algorithm incorporates the idea of optimism in a much more natural way than the related optimistic ADAM algorithm previously proposed by Daskalakis et al. (2018) does. The experiments also indicate some advantage of optimism in the studied optimization problems.

On the other hand, the theoretical contribution is marginal: the algorithm and its analysis is a straightforward combination of previous ideas and the result itself doesn't strike me as surprising at all. Then again, perhaps this is more of a presentation issue, as it may be the case that the authors did not manage to highlight clearly enough the technical challenges they needed to overcome to prove their theoretical results. Furthermore, I find the method for guessing the gradients to be rather arbitrary and poorly explained---at least I'm not sure if anyone unfamiliar with the mentioned gradient extrapolation methods would find this approach to be sensible at all.

I am not fully convinced by the experimental results either, since I have an impression that the gradient-guessing method only introduces yet another knob to turn when tuning the hyperparameters, and it's not clear at all that this new dimension would indeed unlock levels of performance that were not attainable before. Indeed, the authors seem to fix all hyperparameters across all experiments and only switch around the optimistic components, rather than finding the best tuning for each individual algorithm and comparing the respective results. Also, I don't really see any qualitative improvements in the learning curves due to the new components---but maybe I just can't read these graphs properly since I have more of a "theorist" background.

The writing is mostly OK, although there is room for improvement in terms of English use (especially on the front of articles which seem to be off in almost every sentence).

Overall, I don't feel very comfortable about suggesting acceptance, mostly because I find the results to be rather unsurprising. I suggest that the authors try to convince me of the nontrivial challenges arising in the analysis, or about the definite practical advantage that optimism can buy for large-scale optimization.

Detailed comments
=================
- pp.1, abstract: "We consider new variants of optimization algorithms."---This sentence is rather vague and generic. I guess you wanted to refer to *convex* optimization algorithms, which is actually what you consider in the paper. No need to be embarrassed about assuming convexity...
- pp.1: A general nuisance with the typesetting that already shows on the first page is that italic and small capital fonts are used excessively and without any clearly identifiable logic. Please simplify.
- pp.1: "AdaGrad [...] exploits the geometry of the data and performs informative update"---this makes it sound like other algorithms make non-informative updates.
- pp.1: Regret was not defined even informally in the introduction, yet already some regret bounds are compared, highlighting that one "can be much smaller than O(\sqrt{T})". This is not very friendly for readers with no prior experience in online learning.
- pp.1: "Their regret analysis are the regret analysis in online learning."---What is this sentence trying to say?
- pp.2: For this discussion of FTRL, it would be useful to remark that this algorithm really only makes sense if the loss function is convex. Also related to this discussion: you mention that the bound for optimistic FTRL can be much smaller than \sqrt{T}, but never actually say that \sqrt{T} is minimax optimal---without this piece of context, this statement has little value.
- pp.3: "ADAM [...] does not converge to some specific convex functions."---I guess I understand what this sentence is trying to say, but it certainly doesn't say it right. (Why would an *algorithm* converge to a *function*?)
- pp.3, bottom: This description of "extrapolation methods" is utterly cryptic. What is x_t here? What is the "fixed point x^*"? Why is this scheme applicable at all here? (Why would one believe the errors to be near-linear in this case? Would this argument work at all for non-convex objectives?)
- pp.5, Lemma 1: Why would one expect D_\infty to be finite? In order to ensure this, one would need to project the iterates to a compact set.
- pp.5, right after lemma 1: "it does matter how g_t is generated" -> "it does *not* matter how g_t is generated"
- pp.6, top: "we claimed that it is smaller than O(\sqrt{T}) so that we are good here"---where exactly did this claim appear, and in what sense "are we good here"? Also, the norms in this paragraph should be squared.
- pp.6, Sec 3.2: While this section makes some interesting points, its tone feels a bit too apologetic. E.g., saying that "[you] are aware of" a previous algorithm that's similar to yours and doubling down on the claim that "the goals are different" makes the text feel like you're taking a defensive stance even though I can't see a clear reason for this. In my book, the approach you propose is clearly different and more natural for the purpose of your study.

---

> ### Author Response · Authors · 2018-11-27
> **Response to AnonReviewer 1**
>
> Thank you for the valuable comments. We have fixed most of the issued you raised under "Detailed Comments". For now, we leave the small capital letter typesettings for now but we will be happy to remove the capitalization later.
>
> == Regarding to $D_{\infty}$:
> We assume that it is finite. If $w^*$ is finite, then $D_{\infty}$ should be finite. We think it is a reasonable assumption. We will consider the constraint case later.
>
> == Regarding to the extrapolation method:
> We want to use the past few gradient vectors to predict $g_{t}$. So, $x^*$ is the fixed point of the (4).
> If $x_{t-1}= x^{*}$, then $x_{t} = x^{*}$. By using the extrapolation method, we basically assumes that the relation of gradient vectors satisfies (4).
> We admit that it is not true in general but the method helps to get a faster convergence in the experiments.
>
> == Experiment ==
> We think that our results illustrate an obvious acceleration, especially in first few epochs. Since our work focuses on a "optimistic" modification to the AMSGrad algorithm, we choose to compare the improvement brought by different optimistic term based on tuned AMSGrad optimizer, which might be more convincing for the "extra effect".
>
> Again, thanks for the detailed comments.

---

> > ### Comment · AnonReviewer1 · 2018-12-02
> > **response**
> >
> > Thanks for taking your time to respond. Unfortunately, I don't find the responses to be very satisfying.
> >
> > Re boundedness of weights:
> > This is actually a very strong assumption as it implicitly requires a certain kind of stability that ensures that the weights output by the algorithm never diverge. This is already not trivial to ensure when the objective is convex, and certainly much harder if it is non-convex. So, projections are absolutely necessary for this part of the analysis to work out.
> >
> > Re extrapolation method:
> > I do understand that you want to approximate the gradient as the fixed point of (4), I am just having trouble understanding why this would be a reasonable approximation. Specifically, in what cases do you expect this approximation to be accurate? Is there a natural case where the gradients actually follow such a linear recursion? (Maybe linear regression?) What are cases where this method leads to a bad approximation?
> >
> > Please consider discussing these issues in more detail in a future version.

---

### Official Review · AnonReviewer2 · 2018-11-05
**Simple but nice extension with existing ideas from the literature**

**Rating:** 6
**Confidence:** 2

**Review:**

In this manuscript, the authors borrow the idea of "optimism" from the online learning literature and apply it to two frequently used methods for neural network training (AMSGrad and ADAM). More or less, replicating the theory known in the literature, they give a regret analysis. The manuscript ends with a comparison of the optimistic methods against their plain counterparts on a set of test problems.

This is a well-written paper filling a gap in the literature. Through the contribution does not seem significant, the results do support that such extensions should be out there. In addition to a few typos, some clarification on several points could be quite useful:

1) It is not clear why the authors use this particular extrapolation algorithm?

2) If we have the past r+1 gradients, can we put them into use for scaling the next direction like in quasi-Newton methods?

3) The following part of the sentence is not clear: "... the gradient vectors at a specific time span is assumed to be captured by (5)."

4) \nabla is missing at the end of the line right after equation (6).

5) The second line after Lemma 2 should be "... it does not matter how..." (The word 'not' is missing.)

---

> ### Author Response · Authors · 2018-11-27
> **Response to AnonReviewer2**
>
> Thank you for the comments and identifying the typos. We have fixed them. Please find as follows our response.
> We've updated the new version accordingly.
>
> == The extrapolation algorithm ==
> We choose the particular algorithm by (Scieur et al. 2016) because it has good empirical performance
> in practice. (Scieur et al. 2016) shows that using the last few updates of an optimization algorithm,
> the method can predict a point that is much closer to the optimum than the last update of the optimization algorithm.
>
> == Scaling the next direction ==
> This may be a good idea. We leave it as a future work.
>
> == "... the gradient vectors at a specific time span is assumed to be captured by (5)." ==
> We elaborate it in the new version accordingly.
> We want to have a good prediction $m_{t}$ of $g_{t}$ by using the past few gradients.
> If the past few gradients can be modeled by the equation approximately, then
> the method should predict the gradient well.

---

### Official Review · AnonReviewer4 · 2018-11-12
**An interesting way to combine regularized approximate minimal polynomial extrapolation and optimistic online learning.**

**Rating:** 5
**Confidence:** 4

**Review:**

This paper provides an interesting way to combine regularized approximate minimal polynomial extrapolation and optimistic methods. I like the idea and the experimental results look promising. However, I have some concerns:
    - I'm wondering if the comparison with the baseline is fair. Actually, one iteration of Optimistic-AMSGrad is more expensive than one iteration of AMSGrad since it requires to compute m_{t+1}. The authors should explain to what extend this computation is significantly cheaper that a backprop (if it actually is).
    - The discussion after Theorem 1 is not clear. To me it is not clear whether or not Optimistic-AMSGrad has a better Regret that AMSGrad: you did not compare the *sum* of the two additional term with the term with a $\log(T)$ (second line of (8) with second lien of (9)). Do you get a better regret that O(\sqrt{T}), a better constant ? Moreover you did not justify why it is reasonable to assume that each $g_t[i]^2-h_t[i]^2/\sqrt{v_{t-1}[i]}$ are bounded.
    - I'm also concerned about the definition of $D_{\infty}$. Did you prove that this constant is not infinite ? (Reddi et al. 2018) proposed a projected version of their algorithm and did the analysis assuming that the constraints set was bounded. In your Algorithm 2 would you project in Line 8 and 9 or only on line 9 ? I think that the easiest fix would be to provide a projected version of your algorithm and to do your analysis with the standard assumption that the constraints set is bounded.
    - The description of Alg 2 is not clear. "Notice that the gradient vector is computed at w_t instead of w_{t−1/2}" why would it be $w_{t-1/2}$ ? in comparison to what ? "Also, w_{t+ 1/2} is updated from {w_{t− 1/2} instead of w_t." Same. The comments are made without any description of the algorithm, fact is, if the reader is not familiar with the algorithm (which is introduced in the following page) the whole paragraph is hard to catch.
    - Actually, Page 6 you explain how the optimistic step of Daskalikis et al. (2018) is unclear but you can merge the updates Lines 8
and 9 to $w_{t+1} = w_{t} - \eta_{t+1} \frac{4 h_{t+1}}{(1-\beta_1) \sqrt{v_t}} - \eta_t \±rac{\theta_t}{\sqrt{v_t}} + \eta_t \frac{4 h_{t}}{(1-\beta_1) \sqrt{v_{t-1}}}$ (Plug line 8 in line 9 and then plug Line 9 at time t) to get a very similar update. If you look more closely at Daskalakis et al. 2018 their guess $m_{t+1}$ is $g_t$. Finally you Theorem 2 is stated in a bit unfair way since you also require $\beta_1 <\sqrt{\beta_2}$, moreover it seems that Theorem 2 is no longer true anymore if, as you says, you impose that the second moment of ADAM-DISZ is monotone adding the maximization step.
    - About the experiments, I do not understand why there is the number of iterations in the left plots and the number of epochs on the right plots. It makes the plots hard to compare.
    - You should compare your method to extragradient methods.



To sum up, this paper introduce interesting results. The combination of (Scieur et al. 2016) and optimistic online learning is really promising and solid theoretical results are claimed. However, some points should be clarified (see my comments above). Especially, I think that the authors focused too much (sometimes being unfair in their discussion and propositions) on showing how their algorithm is better than (Daskalakis et al. 2018) whereas as they mentioned it "The goals are different." ADAM-DISZ is designed to optimize games and is similar to extragradient. It is know that extragradient methods are slower than gradient methods for single objective minimization because of the extrapolation step using a too conservative signal for single objective minimization.


Some minor remarks:
    - Page One "NESTEROV'SMETHOD"
    -  "which can be much smaller than \sqrt{T} of FTRL if one has a good guess." You could refer to Section 3 or something else because otherwise this sentence remains mysterious. What is a good guess (OR maybe you could say that standard "good" guesses are either the previous gradient or the average of the previous gradients)
    - "It combines the idea of ADAGRAD (Duchi et al. (2011)), which has individual learning rate for different dimensions."  what is the other thing combined ?
    - Beginning of Sec 3.1 $\psi_t$ represent $diag(v_t)^{1/2}$.

===== After Authors Response =====
As developed in my response "On $D_{\infty}$ assumption " to the reviewers, I think that the assumption that $D_\infty$ bounded is a critical issue.
That is why I am moving down my grade.

---

> ### Author Response · Authors · 2018-11-27
> **Response to AnonReviewer4**
>
> Thank you for the valuable comments. Please find as follows our response.
> We've updated the new version accordingly.
>
> == Iteration Cost ==
> Yes, like optimistic algorithms in general, each iteration would be slightly more expensive.  Using fewer iterations means that it has fewer access to training samples than what required by AMSGrad. The sample efficiency might have some advantages on certain applications. One can also design some algorithms or update rules to reduce the time of predicting $m_t$.
>
> == Theorem 1 ==
> Yes, we should also compare the sum of the last two terms in Theorem 1 (namely, (8)) with the last term on (9). We are not claiming that the regret bound of Theorem 1 is always better than that of AMSGrad (namely, (9)). Our original discussion actually means that if $m_{t}$ and $g_{t}$ is close, then the last term of (8) would dominate. We then try to upper-bound it in a very loose sense (i.e. (10)). We treat each $\frac{(g_t[i]^2- h_t[i]^2)}{\sqrt{\hat{v}_{t-1}[i]}}$ as a constant and get a $O(\sqrt{T})$ bound. However, in practice, $\sqrt{\hat{v}_{t-1}[i]}$ might grow over time,
> and the growth rate is different for different dimension $i$. If $\sqrt{\hat{v}_{t-1}[i]}$ grows like $O(\sqrt{t})$ then the last term of (8) is just $O(\log T)$, which might be better than that of the last term on (9). One can also get a data dependent bound like the last term on (9). We just wanted to say that the regret bound might be better than that of AMSGrad under certain conditions.
>
> == $D_{\infty}$ ==
> We assume that it is finite. If $w^*$ is finite, then $D_{\infty}$ should be finite. We think it is a reasonable assumption. Thanks for your suggestion, we will consider the constraint case later.
>
> == Discussion/Description of Algorithm 2 ==
> Thanks for the suggestion. We updated it accordingly.
>
> == Comparison with the update of ADAM-DISZ on Page 6 ==
> Thanks for the suggestion. We updated the discussion accordingly.
> In short, combining (8) and (9) in Algorithm 2, we get that
> $$ w_{t+1} = w_{t-1/2} - \eta_t \frac{\theta_t}{\sqrt{\hat{v}_t}}
> - \eta_{t+1} \frac{4}{1-\beta_1} \frac{h_{t+1}}{\sqrt{\hat{v}_t}}
> $$ Notice that,  $w_{t+1}$ is updated from $w_{{t-1/2}}$ instead of $w_{{t}}$. So, ADAM-DISZ is not really similar to the proposed algorithm here.
>
> == Experiments ==
> On the left we want to show the whole training path, and we followed the choice of axis units as in previous related works (e.g Reddi et al. 2018). On the right we want to focus on the acceleration in relatively early stage, so we choose to plot against number of epochs to highlight this point. We have added some explanation in the caption of figure 1. Thanks for your suggestion.

---

> > ### Comment · AnonReviewer4 · 2018-12-04
> > **On $D_{\infty}$ assumption**
> >
> > I think that assuming that $D_\infty$ is bounded is a lack of rigor. It is not a property of the problem but a property of the algorithm you use.  The goal is to show the appealing properties of your algorithm.
> >
> > Let me give you an example. if we consider the simple sequence $x_{t+1} = x_t - \eta x_t$ (which is basically gradient descent on the one dimensional objective $f(x) = x^2$. ) Then if we assume that $D_{\infty} = sum_{t} \|x_t\|$ is bounded we have:
> >
> > $ \eta x_{t} = x_t - x_{t+1} => \frac{\eta}{T} \sum_{t=0}^T x_t = (x_0 - x_{T+1})/T
> >                                           => \eta \|\frac{1}{T} \sum_{t=0}^T x_t \| \leq  2 D_\infty/T$
> >
> > Now with $\eta = T$  we get that the average of $x_t$  converge at a rate $O(1/T^2)$.
> > $\|\frac{1}{T} \sum_{t=0}^T x_t \| \leq  2 D_\infty/T^2$
> >
> > That is not true because for $\eta > 1$ it is easy to show that this sequence actually diverge !
> >
> > This example underlines that $D_{\infty}$ may actually depend on $\eta$.
> >
> > I consider that it is paradoxical to assume that a sequence is bounded in order to eventually show that it's average actually converge.

---

### Meta-Review · Area_Chair1 · 2018-12-18
**Borderline paper**

**Confidence:** 4
**Recommendation:** Reject

**Metareview:**

The reviewers expressed some interest in this paper, but overall were lukewarm about its contributions. R4 raises a fundamental issue with the presentation of the analysis (see the D_infty assumption). The AC thus goes for a "revise and resubmit".